# Computational modelling of muscle fibre operating ranges in the hindlimb of a small ground bird (*Eudromia elegans*), with implications for modelling locomotion in extinct species

**Peter J. Bishop**[1,2¤]*, **Krijn B. Michel**[1], **Antoine Falisse**[3,4], **Andrew R. Cuff**[1,5], **Vivian R. Allen**[1], **Friedl De Groote**[3], **John R. Hutchinson**[1]

**1** Structure and Motion Laboratory, Department of Comparative Biomedical Sciences, Royal Veterinary College, Hatfield, United Kingdom, **2** Geosciences Program, Queensland Museum, Brisbane, Australia, **3** Department of Movement Sciences, KU Leuven, Leuven, Belgium, **4** Department of Bioengineering, Stanford University, Stanford, California, United States of America, **5** Hull York Medical School, University of York, York, United Kingdom

¤ Current address: Museum of Comparative Zoology, Department of Organismic and Evolutionary Biology, Harvard University, Cambridge, Massachusetts, United States of America
* pbishop@fas.harvard.edu

**Data Availability Statement:** The authors confirm that all data underlying the findings are fully

## Abstract

The arrangement and physiology of muscle fibres can strongly influence musculoskeletal function and whole-organismal performance. However, experimental investigation of muscle function during *in vivo* activity is typically limited to relatively few muscles in a given system. Computational models and simulations of the musculoskeletal system can partly overcome these limitations, by exploring the dynamics of muscles, tendons and other tissues in a robust and quantitative fashion. Here, a high-fidelity, 26-degree-of-freedom musculoskeletal model was developed of the hindlimb of a small ground bird, the elegant-crested tinamou (*Eudromia elegans*, ~550 g), including all the major muscles of the limb (36 actuators per leg). The model was integrated with biplanar fluoroscopy (XROMM) and forceplate data for walking and running, where dynamic optimization was used to estimate muscle excitations and fibre length changes throughout both gaits. Following this, a series of static simulations over the total range of physiological limb postures were performed, to circumscribe the bounds of possible variation in fibre length. During gait, fibre lengths for all muscles remained between 0.5 to 1.21 times optimal fibre length, but operated mostly on the ascending limb and plateau of the active force-length curve, a result that parallels previous experimental findings for birds, humans and other species. However, the ranges of fibre length varied considerably among individual muscles, especially when considered across the total possible range of joint excursion. Net length change of muscle–tendon units was mostly less than optimal fibre length, sometimes markedly so, suggesting that approaches that use muscle–tendon length change to estimate optimal fibre length in extinct species are likely underestimating this important parameter for many muscles. The results of this study clarify and broaden

available without restriction. Calibration and undistortion images and X-ray video files used in this study are available through the X-ray Motion Analysis Research Portal (http://xmaportal.org/, study identifier 'Tinamou Walking and Running'). Model and code files are provided in the Supporting Information.

**Funding:** This work was supported by the ERC Horizon 2020 Advanced Investigator Grant (695517, to J.R.H.; https://erc.europa.eu/), and Research Foundation Flanders (grant number G079216N, to F.D.G.; https://www.fwo.be/). The funders had no role in study design, data collection and analysis, decision to publish, or preparation of the manuscript.

**Competing interests:** The authors have declared that no competing interests exist.

understanding of muscle function in extant animals, and can help refine approaches used to study extinct species.

## Author summary

The structure and behaviour of individual muscles greatly influence an animal's ability to move and perform tasks. Experimental investigation of how muscles function within a living animal can be difficult, providing insight on limited aspects for few muscles. Computer models of the musculoskeletal system can partly overcome this challenge, and provide valuable insight into features not easily (if at all) measurable in experiments alone. Here a detailed three-dimensional musculoskeletal model of the hindlimb of a small ground bird, a tinamou, is combined with experimental motion and force data for walking and running, and used to explore muscle function during these gaits. Feeding experimental data into the model, a simulation is used to estimate muscle activity, and the pattern of change in their constituent fibres. Fibre length change is an important determinant of muscle function because the amount of force that can be produced varies with the amount and rate of contraction or stretch. For the first time, patterns of fibre length change in every key muscle of the bird hindlimb have been studied, broadening understanding of muscle function across the diversity of modern animals. The results also have bearing on how muscle function is reconstructed for extinct animals (e.g., dinosaurs).

## Introduction

The ultimate driver of almost all vertebrate movement is skeletal muscle [1,2]. Spanning from one bone to another, the contractile force of a muscle–often delivered via in-series tendon–exerts a moment (rotational force) about one or more joints, effecting movement. Whilst the basic function of muscle had been recognized at least since Galen of the second century A.D. [3], only in the past half a century has the highly complex behaviour of skeletal muscle and tendon in relation to whole-organism biomechanics become increasingly appreciated. Much of this advancement in understanding has been due to the use of various forms of lumped-parameter models based on the pioneering work of Hill [4], which emphasize the contributions of both active and passive components to force production, and how these are modulated by the degree and rate of muscle shortening or lengthening [1,5–7]. Empirical data on the biomechanics of muscles *in vivo* [e.g., 8,9] and *ex vivo* [e.g., 10,11–13] form the foundation of this understanding. However, experimentally investigating muscle function during *in vivo* activity is difficult even for a single large muscle, and is practically infeasible for understanding the function of every muscle in a given system (e.g., a limb, which may comprise in excess of 40 muscles). Computational models of the musculoskeletal system using Hill-type models of muscle contraction have therefore proven as valuable tools to investigating the complex, highly dimensional and highly nonlinear ways in which muscles help coordinate steady and unsteady movement in various species [14–18], and how these can vary in abnormal conditions such as pathology [19–22].

Despite great advancements in understanding muscle–tendon function insofar as it relates to whole-organism performance and behaviour, the majority of research effort has focused on a single species, humans. Yet, the diversity of form and behaviours exhibited by other species can provide an enriched insight into the relationships between anatomy, function and whole-

organism performance [2], and moreover can provide motivation for bioinspired assistive and robotic technologies [23–25]. Bird species, particularly ground-dwelling ones, have received considerable attention due to their habitual, parasagittal gait and bipedal posture [26–28,29, and references cited therein]. Notwithstanding important anatomical, kinematic and kinetic differences from humans, terrestrial locomotion in birds has been frequently studied as a model system for bipedal locomotion. This has provided enriched insight into diverse topics including energetics [30–33], performance [34,35], stability [36,37], neuromuscular control [27,33,38] and the behaviour of muscles with unusual anatomies (including long tendons) [36,38,39], as well as how these aspects vary with increasing body size or speed, or under different environmental conditions [32,40–44]. However, as with humans or any other species, such experimental research is frequently limited in the number or diversity of muscles that can be practically investigated in any given subject or study. This is further compounded by the smaller body size of almost all bird species and the practical nuances of experimental work with animals.

Computational models of the musculoskeletal system again can help provide unique perspectives on aspects of musculoskeletal function that are not easily (if at all) measureable in experiments. A small number of studies have applied musculoskeletal modelling to avian terrestrial locomotor biomechanics and hindlimb function [e.g., 17,40,45–50]. In association with different aims, these studies have varied in their level of sophistication and anatomical scope, ranging in focus from a single joint through to the whole limb. Consequently, whilst muscles been classified as to their broad role in the avian hindlimb during locomotion [17], many key aspects of muscle–tendon behaviour remain underexplored.

One such aspect is muscle fibre length, how this varies across different muscles and how this varies throughout different dynamic behaviours. (In this paper, 'fibre length' refers to the common usage of the term as it pertains to musculoskeletal modelling, and anatomically most closely corresponds to muscle fascicles rather than the microscopic fibres themselves [5].) As encapsulated by Hill-type models, a muscle's force-producing capacity greatly varies depending on how stretched or contracted its fibres are, as well as how quickly they stretch or contract. Not surprisingly, sensitivity analyses of musculoskeletal models of locomotion and other behaviours demonstrate the important effect that variance in optimal fibre length ($\ell_o$, the length at which isometric force is maximal) can have on muscle–tendon function [51,52], patterns of muscle recruitment [53] and whole-organism performance [45,54]. Fibre length changes during avian locomotion have been experimentally investigated previously [e.g., 9,36,38,41,55,56–58] but such studies have necessarily focused on a select few muscles and species. An understanding of fibre length variation in all hindlimb muscles during locomotion would therefore provide clarity on the diversity of muscle functions, and lead to a more developed comprehension of the relationship between muscle–tendon anatomy and whole-organism biomechanics.

The relevance of fibre length to understanding organismal function, performance and ecology extends beyond just extant species. Attempts to better understand the behaviour and function of enigmatic extinct species, such as non-avian dinosaurs [54,59–68], early hominids [69–71] or mammal ancestors [72], have often employed musculoskeletal models of varying complexity. A central issue for each of these studies is the fact that various biomechanically relevant parameters remain unknown for fossil species, especially aspects pertaining to soft tissues such as muscle size (strength) and $\ell_o$. These parameters must therefore be estimated. The approaches that have been used to estimate soft tissue parameters have varied widely, ranging from comparison to empirical datasets derived from extant species [e.g., 52] through to geometric assumptions based on skeletal proportions [60]. In terms of $\ell_o$, one approach to estimation strives to incorporate as much information as possible from the fossil bones, using both

bone dimensions and joint range of motion [66,73,74]. This approach assumes that across a joint's entire range of motion, a given muscle's fibres will undergo length change of $0.5–1.5\times$ $\ell_o$ (which is the range over which a fibre can actively develop force in many Hill-type models [5]), in turn allowing $\ell_o$ to be calculated from measured joint range of motion and estimated moment arm. In essence, the approach assumes that muscle fibre lengths are 'tuned' (i.e., structurally optimized) to traverse their full active force–length curve across a limb's range of motion. This assumption remains to be quantitatively tested, and also contrasts with the findings of numerous studies that have demonstrated that muscle fibres tend to operate over a more restricted range of lengths during physiological activity, largely on the ascending limb or plateau of the active force–length curve [e.g., 8,15,41,51,58,75,76–81]. Moreover, for some muscles at least, tendon compliance and biarticularity can decouple change in fibre length from muscle moment arm and change in joint angle [50,82], further complicating the issue.

The principal aim of this study was to use computational methods to explore muscle excitations and fibre length changes during avian gait. These results would help test whether fibres in the majority of muscles in the avian hindlimb do remain on the ascending limb and plateau of their active force–length curve, as suggested from more limited experimental observations in birds (and other species). Experimental locomotion data were synthesized with a three-dimensional (3-D) computational musculoskeletal model of the hindlimb of a small, generalized ground bird, the elegant-crested tinamou (*Eudromia elegans*). This is the first time that high-quality fluoroscopic kinematic data have been integrated with a complete musculoskeletal model of avian terrestrial locomotion. An inverse simulation employing dynamic optimization was used to estimate muscle excitations and fibre length changes throughout the walking and running stride for almost all the key muscles of the hindlimb, the first time that this has been attempted for a bird. As palaeognaths, tinamous presumably closely represent ancestral avian form and function [83,84], providing a contrast with prior experimental studies that have investigated neognath species, which may have more derived musculoskeletal anatomies or functions. Moreover, this also provides a contrast with the only other full-limb inverse study of muscle function in bird locomotion, which was conducted on the ostrich [17], a species more than two orders of magnitude greater in size. Stemming from the results of the inverse simulation, a secondary aim of this study was to use a series of static simulations to further explore how broadly fibre length varies across the whole range of possible limb postures (of which only a subset is used in *in vivo* locomotor behaviour). Collectively, the results of these simulations provide the first test of the above approach used in studies of extinct species (i.e., fibre length changes encompass $0.5–1.5\times \ell_o$), and provide important new insight into muscle function in non-human species.

## Materials and methods

### Ethics statement

All experimental protocols were conducted in the Structure and Motion Laboratory of the Royal Veterinary College, via prior approval by the College's Ethics and Welfare Committee (approval number 2016-0089N) and under a project licence (P0806ABAD) granted by the Home Office (United Kingdom).

### Study overview

This study can be divided into six main phases (Fig 1), for which a fully detailed description is given in the sections below. Briefly, kinematic and kinetic data were first collected for tinamous walking and running at a range of speeds, using both overground and treadmill experimental setups. Secondly, the raw experimental data was processed. Thirdly, a high-fidelity,

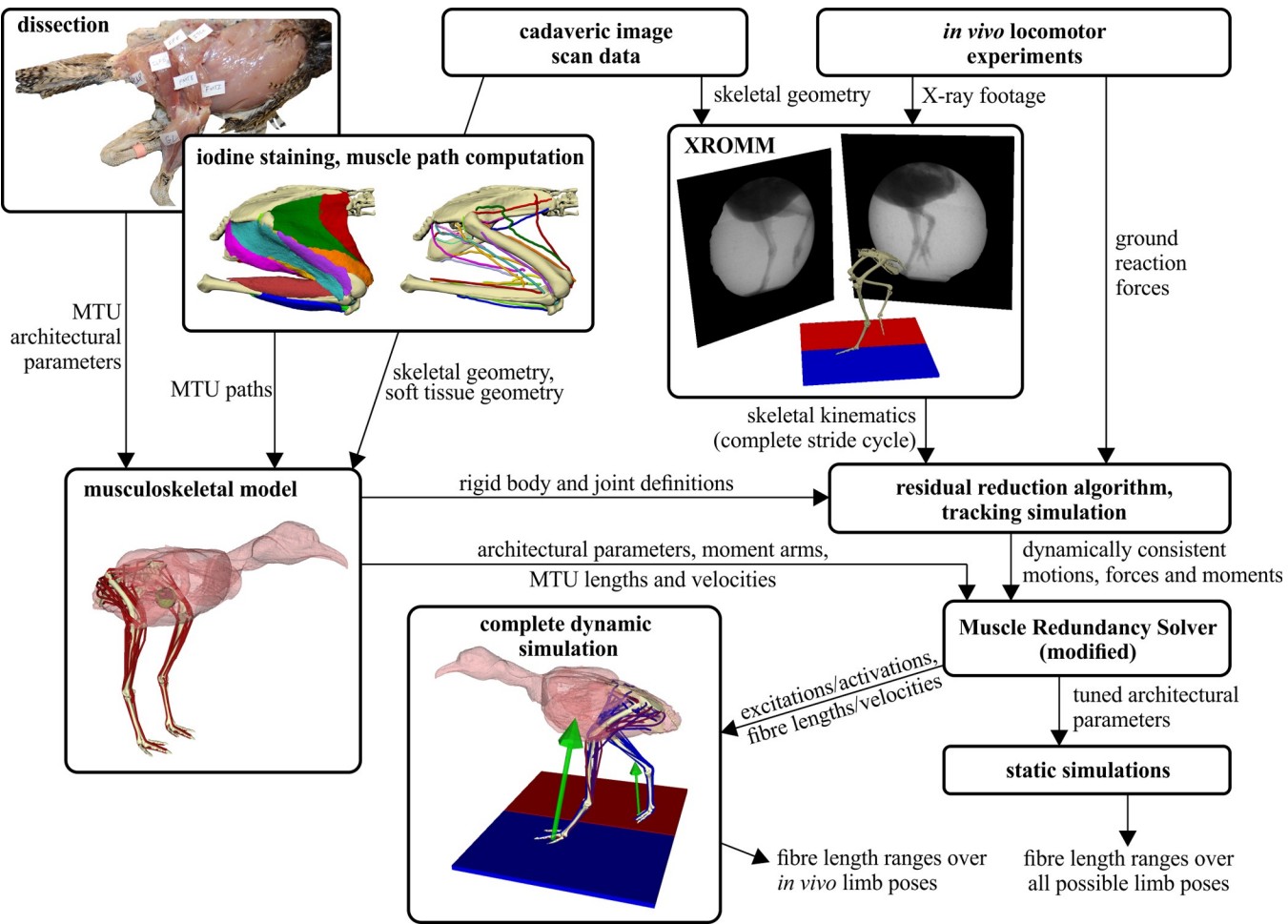

**Fig 1. Schematic overview summarizing the workflow of the study.** The input data are anatomical measurements of muscles and bones (obtained by physical and digital means) and *in vivo* experimental data; the outputs are a high-fidelity computational musculoskeletal model, estimations of *in vivo* muscle function, and estimations of the range of viable operating lengths of each muscle. XROMM, X-ray reconstruction of moving morphology; MTU, muscle–tendon unit.

3-D musculoskeletal model of one of the tinamou subjects was developed, incorporating subject-specific anatomical data. Fourthly, the processed experimental data for the slowest walk and fastest run were synthesized with the musculoskeletal model, in order to capture the greatest possible variation in steady-state locomotor behaviour across the recorded data. Dynamic inconsistency between kinematics, kinetics and the model was rectified by using a residual reduction algorithm and a tracking simulation. Fifthly, an inverse simulation was conducted, where external joint moments (computed as an inverse dynamics output from the tracking simulation) were balanced by the muscles. A modified implementation of the dynamic optimization routine of De Groote et al. [7] was used here, wherein the walking and running trials were both solved for whilst simultaneously tuning optimal fibre length and tendon slack length for all muscles. This procedure estimated time histories of muscle excitations and fibre length changes for both gaits, and also derived tuned architectural parameters for each muscle in the model. Lastly, the tuned musculoskeletal model was subject to four sets of static simulations to elicit the full range of fibre lengths achievable across the entire range of limb postures, only a subset of which were actually used *in vivo*, thus better delimiting the true extent of viable operating fibre lengths.

## Experimental data collection

Four birds contributed data in various forms to the current study (Table 1). Although additional individuals were used in the experiments, logistical and experimental constraints meant that only two (DDT04 and DDT09, both of near-identical mass and linear dimensions) contributed *in vivo* data to the study.

Kinematic data were collected via biplanar fluoroscopy, using a combination of marker-based and markerless X-ray reconstruction of moving morphology (XROMM [85,86]). Two BV Libra C-arm systems (Koninklijke Philips N.V., Amsterdam, Netherlands) were used, each comprising a BV 300 generator, F017 tube and BV 300 collimator and intensifier (22.9 cm diameter), with a source-to-image distance of 99.5 cm. These were backed by custom-mounted Photron FASTCAM Mini WX50 high-speed digital video cameras (Photron, Tokyo, Japan), recording at 750 frames/second with a 1/750 s shutter speed and 2048×2048 pixel resolution. Prior to the commencement of experiments each day, still images of perforated sheet metal and a custom-made calibration structure were captured to facilitate image undistortion and 3-D calibration of the camera positions in the experimental setup [86,87]. The calibration structure consisted of 64 radio-opaque 2.0 mm steel ball bearings set in a cubic grid constructed from radio-transparent acrylic, the dimensions of which were precisely known. To facilitate reconstruction of bone motions, radio-opaque 1.0 mm spherical tantalum beads (Bal-Tec, Los Angeles, USA) were surgically implanted into the birds' bones via press-fitting into hand-drilled holes; surgical and anaesthetic techniques are described by Cuff et al. [88] and Ronaldson et al. [89]. The small size and gracility of the bones, in addition to welfare considerations for the birds, limited the number and placement of the beads: eight in total for DDT04 (two in the pelvis, one in the femur, two in the tibiotarsus and three in the tarsometatarsus) and six for DDT09 (three each in the right tibiotarsus and tarsometatarsus). Birds were given six days to recover from surgery prior to the commencement of experiments; qualitative observations indicated no apparent lameness or otherwise different locomotor behaviour in the birds following surgery.

Two experimental setups were used to collect locomotion data, one overground, the other using a treadmill. The overground setup consisted of a 244×38 cm runway walled with plywood and acrylic (wall height 40 cm), with two Kistler Z17097 forceplates (Kistler Instruments Ltd, London, UK; 20×10 cm each) mounted side-by-side in the middle, set flush with the surrounding runway surface. To reduce slippage, the runway surface was covered with thin rubber matting and the forceplates' top plates covered with anti-slip tape. The forceplates were operated through two Kistler model 9865 amplifiers, which provided analogue inputs to a custom-built data acquisition device (DAQ) that converted voltages to a digital signal; this signal was recorded using a custom LabVIEW 2017 (National Instruments, Austin, USA) script, sampling at 500 Hz. The X-ray systems were positioned straddling either side of the runway, oriented in approximately anterolateral and posterolateral views. The treadmill setup consisted of a 100×40 cm dog treadmill (Starkerhund, Terraglione di Vigodarzere, Italy) that was walled by

**Table 1. Checklist of tinamou individuals that contributed data in the study.**

| Individual | Mass (g) | Data contributed | | | | | |
| --- | --- | --- | --- | --- | --- | --- | --- |
| | | Skeletal | Muscle architecture | Muscle paths | Segment mass properties | Kinematics | Kinetics |
| DDT03 | 524 | | | X | | | |
| DDT04 | 540 | X | | | | X | |
| DDT07 | 595 | | | | X | | |
| DDT09 | 534 | X | X | | | X | X |

acrylic, with the X-ray systems oriented in approximately lateral and dorsal views. Ground reaction forces (GRFs) and moments (GRMs) were not measured in this setup. The cameras (and forceplate, in the overground setup, via the DAQ) were synchronized using a manual trigger pulse. In the overground setup, birds either moved on their own accord at a self-selected speed or moved following external motivation, such as making noises. In the treadmill setup, birds started on a static tread, the speed of which was then gradually increased as the birds tried to keep pace; data recording commenced when it was judged that the bird was moving in a steady fashion at a constant tread speed. A wide range of tread speeds was attempted, and birds were allowed to rest for at least one minute between successive trials. Following the conclusion of experiments, the birds were euthanized via cervical dislocation, massed and stored frozen at -20˚C. Subsequently, the still-frozen carcasses were imaged via X-ray micro-computed tomographic (CT) scanning to capture skeletal morphology and XROMM bead placement (Nikon XTEK XTH 225 ST [Nikon Metrology NV, Leuven, Belgium], 200 kV peak tube voltage, 0.2 mA tube current, 708 ms exposure time, 0.092–0.125 mm isotropic voxel resolution) at the University Museum of Zoology (Cambridge, UK).

## Experimental data processing

**Kinematics.** X-ray videos were undistorted, calibrated and digitized using XMALab 1.5.4 [87], which also computed rigid body motions for the bones where possible (i.e., when a bone had three beads implanted in it). In addition, the tip of the ungual of digit III from both limbs was digitized to grossly capture movement at the metatarsophalangeal (MTP) joints. The limited experimental data collected, in terms of both number of trials and number of beads, did not permit an assessment of tracking accuracy or precision [86] for the current dataset. However, in a separate cadaveric study of tinamous (using different individuals and bead placements) a much larger dataset of bone motions was obtained using the same XROMM system as above. Across the 49 trials in this dataset there were 358 pairwise co-osseous intermarker distances recorded over a total of 149,795 frames; the known distances between these markers (measured in the cadaveric CT scans) was used to determine the accuracy of the experimental setup and digitization procedure. In terms of accuracy, there was a mean relative error of 0.1259% (mean absolute error of 0.08146 mm), and in terms of precision, there was a mean standard deviation of 0.06839 mm. Although it remains uncertain how strictly these results translate to the kinematic data of the present study, they at least give a qualitative indication of acceptable levels of accuracy and precision.

From the digitized kinematic data and recorded forceplate data, two trials were identified to be the focus of the inverse simulations on the basis of cleanness of the forceplate data and completeness of kinematic data. Moreover, these trials spanned practically the entire range of speeds that was able to be elicited from the birds–a slow walk (DDT09, 0.39 m/s or a relative speed [90] of approximately 0.32, duty factor 0.71, overground) and a grounded run (DDT04, 1.39 m/s or a relative speed of approximately 1.15, duty factor 0.57, treadmill)–and they therefore capture the full range of variation in locomotor behaviour that was elicited in the birds.

Bone motions in these focal trials were reconstructed from the digitized bead trajectories and rigid body motion output using a combination of markerless and maker-based rotoscoping [85] in Maya 2018 (Autodesk, San Rafael, USA). This involved first generating the geometries of the bones, as well as the location of the XROMM beads in them, which was accomplished via a combination of manual and automatic segmentation of the carcass CT scans using Mimics 20.0 (Materialise NV, Leuven, Belgium), followed by mesh refinement in 3-Matic 12.0 (Materialise NV, Leuven, Belgium). Rotoscoped bone motions in the global (laboratory) coordinate system were then expressed in terms of physiologically meaningful joint

movements using a series of joint coordinate systems, following the approach and convention of Kambic et al. [91]. Briefly, geometric primitives (e.g., spheres, cylinders) were fitted to the articular surfaces of bones using MATLAB code previously described by Bishop et al. [92], and used to mathematically derive anatomical coordinate systems for the proximal and distal ends of each bone. These were then united to produce a joint coordinate system for each joint, using the 'jointAxes' script in the XROMM_MayaTools package (available at www.xromm. org), which computed joint translations and rotations. The fibula and tibia were not permitted any relative motion between them, rather moving as a single entity. This study follows the previously used convention of expressing joint rotations as three intrinsic Euler rotations, measured as flexion–extension, followed by abduction–adduction, followed by long-axis rotation [91,93]. Due to the small size of the calibrated X-ray volume in relation to the size of the birds, the pelvis was not fully visible in both X-ray videos throughout the entirety of the trials. As such, its position and orientation were estimated from the reconstructed positions of the femora, wherein the centres of the femoral heads were used as 'virtual points' to help constrain the pelvis's disposition by assuming that these corresponded as closely as possible to the centres of the acetabulae [cf. 91,94].

An additional consequence of the small size of the calibrated X-ray volume was that neither of the focal trials' data comprised an unbroken kinematic sequence covering a full stride cycle. Missing sections of the kinematic sequences were filled in by making a composite sequence from corresponding sections recorded elsewhere, using instances of foot touchdown and liftoff (i.e., stance and swing) as points by which to bring different strides to a common timescale. The approach inherently involves a level of subjectivity, but the overarching philosophy was to synthesize a complete stride cycle that as faithfully as possible reflected the kinematics of the focal limb or stride involved. A hierarchy of importance was followed in terms of how missing data were sourced. If possible, gaps in the focal limb's kinematics were first filled in using kinematics from the same limb in an earlier or later stride. Then, gaps were filled using kinematics from the contralateral limb in the same trial; this assumed bilateral symmetry in limb movements, which for the trials involved was supported by a comparison of duty factors and phase offsets between left and right limbs [but see 95]. Lastly, gaps were filled in using kinematics from the focal limb, but from a separate trial (or trials) of as similar speed as possible. For the treadmill trial, which encompassed three strides, a missing section of data for a given joint was filled in using data for that joint in a different stride (via cubic spline interpolation in MATLAB), preferably the same joint but otherwise that of the contralateral limb. The overground trial encompassed less than a single stride within the field of view of the X-ray cameras, and missing sections of data for one joint were filled in using data from the other joint, again using cubic spline interpolation. However, some small gaps remained in the kinematic sequence, and these were filled in using data from a separate treadmill trial involving the same bird using a similar slow walk (~0.45 m/s). The completed kinematic sequences were then filtered with a fourth-order, zero-lag, low-pass Butterworth filter; for the walking trial a cutoff frequency of 20 Hz was used, whereas a cutoff frequency of 30 Hz was used for the running trial. Lastly, for the running trial the tread speed was added to the forward translational coordinate of the pelvis to emulate overground running. Ultimately, slightly more than one whole stride was reconstructed for both walking and running. The approach used to derive whole-stride kinematics is admittedly not ideal, and implicitly assumes that kinematics obtained from overground and treadmill setups are equivalent; the observations of Jacobson and Holly-day [96] suggest that this is broadly the case, but more detailed studies may detect consistent finer-level differences [e.g., 97,98]. Nevertheless, the resulting kinematic sequences were qualitatively highly comparable to those previously reported for other small- to medium-sized ground birds, including *Eudromia elegans*, moving at similar relative speeds [26,95,99].

**Kinetics.** As the running trial occurred on a treadmill, no ground reaction data were collected; see below for how this was estimated and used in the simulations. The walking trial fortuitously involved each foot landing cleanly and wholly on adjacent forceplates, such that complete force and moment profiles were recorded. The raw signal from the forceplates was calibrated, baselined and filtered using a custom script in MATLAB (v9.5; MathWorks, Natick, USA), which computed forces, moments and the instantaneous centre of pressure; filtering used a fourth-order, zero-lag, low-pass Butterworth filter, with a cutoff frequency of 15 Hz. A lower cutoff frequency was used here compared to the kinematic data due to differences in the quality of the two data sets; for example, the forceplate data tended to contain more noise, and so a more aggressive filtering was chosen to conservatively keep only the major aspects of data variation. The cutoff frequencies used in the present study are similar to those used in previous studies of avian terrestrial locomotion [28]. However, in future studies that use greater sample sizes than used here, and use higher-quality experimental data, a single consistent cutoff frequency would likely be desirable [100]. Reconstructing the complete GRF and GRM profiles through time for the whole stride involved reconstructing GRFs and GRMs for the period of double stance at the beginning and end of the trials, periods for which forces and moments of only one foot were recorded. This was achieved by duplicating the GRF and GRM profile for the foot involved and shifting it forward or backward in time as appropriate so as to correspond with instances of foot touchdown and liftoff (estimated using equation S4 of [28]). The centre of pressure of these unmeasured footfalls was estimated from the position of the relevant foot involved, using the reconstructed kinematic sequence derived above.

## Musculoskeletal model development

**Skeletal and joint definitions.** A high-fidelity 3-D musculoskeletal model of the pelvis and hindlimbs was developed for use in OpenSim 3.3 [101,102] (Fig 2 and see S1 Model File). It was based principally on individual DDT09, with data on muscle lines of action and segment mass properties drawn from two additional individuals (Table 1). The skeletal geometry and joint definitions of the model were identical to those used above in deriving locomotor kinematics, wherein a bilaterally symmetrical, rigged hierarchical marionette [85] was created in Maya and then transcribed to the OpenSim modelling environment as per the workflow of Bishop et al. [92]. Digits II–IV were modelled together as a single segment. For computational simplicity, joints in the OpenSim model were permitted rotation only (no translations), and moreover the MTP joint was permitted rotation about the flexion–extension axis only. Therefore, each limb comprised 10 degrees of freedom (DOFs), with three each at the hip, knee and ankle, in addition to six describing the location and orientation of the pelvis and trunk segment in the global coordinate system. In the XROMM joint convention followed here, the neutral pose of the joint coordinate systems (i.e., where all joint translations and rotations are zero) involves the bones collapsed into a highly non-physiological concertina (Fig 2B; see also [91]). The neutral pose of the musculoskeletal model was set differently to this by incorporating a fixed offset in all flexion–extension axes (hip extension +90˚, knee extension +180˚, ankle extension +180˚, MTP angle + 90˚), such that the limb was vertically straightened with the digits segment parallel to the ground plane (Fig 2A). By not accounting for joint translations in the model, this inevitably resulted in a degree of interpenetration between adjoining bones. As such, fixed translational offsets were added to definitions of the knee (5 mm), ankle (4 mm) and MTP joints (3.5 mm) in order to avoid interpenetration. The displacements were applied only along the proximodistal axis of the 'child' body (e.g., the tarsometatarsus for the ankle joint), and the magnitude of displacements was based on the joint translations measured in the *in vivo* kinematic sequences.

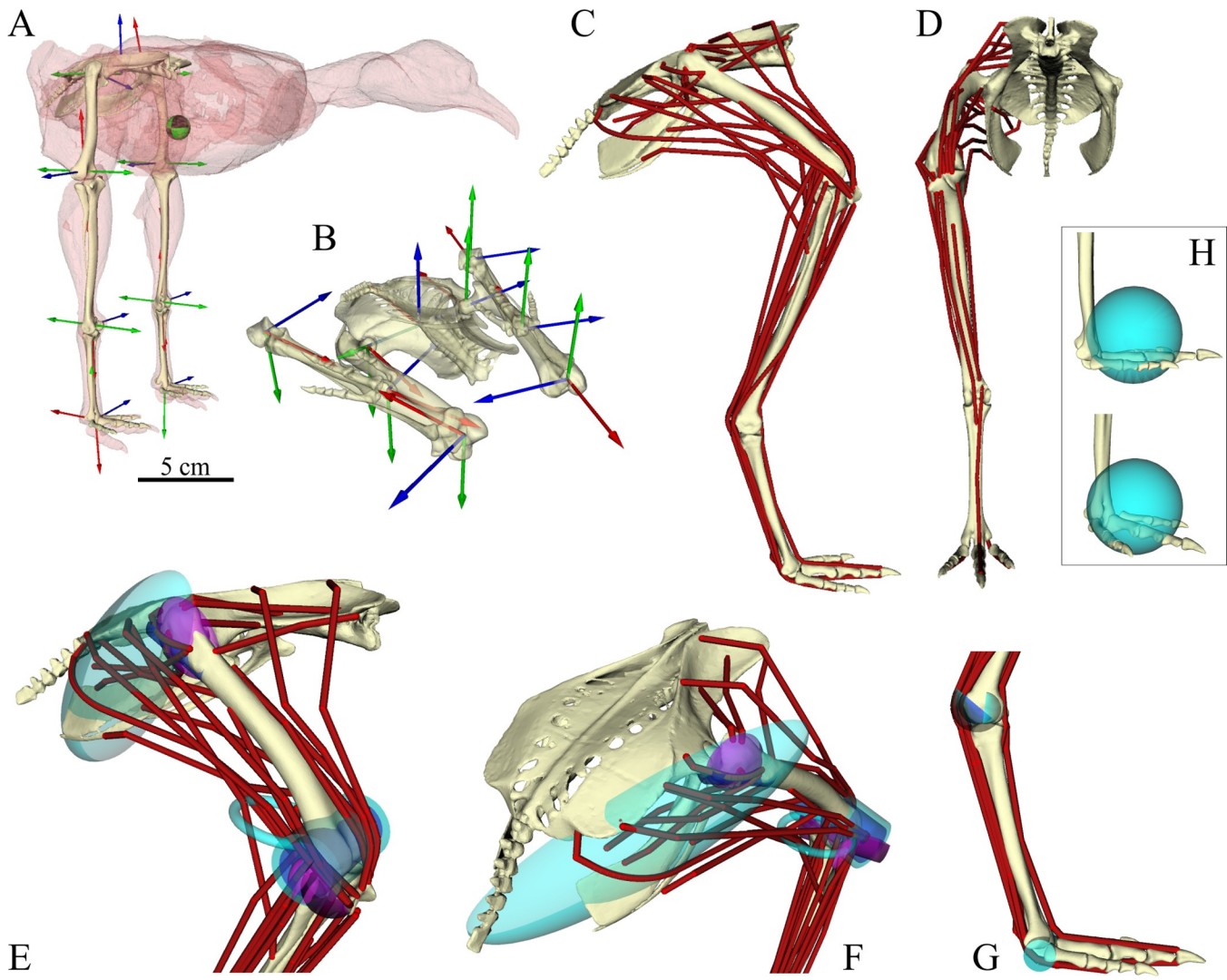

**Fig 2. Musculoskeletal model of the tinamou.** (A) Rigid body mechanics component of the model, shown in the neutral pose used in this study; the green and black sphere denotes the whole-body centre of mass in this pose. Scale bar refers to this panel. (B) Skeleton and joints shown collapsed in the XROMM neutral pose [91], which differs from the neutral pose of the musculoskeletal model by offsets of 90° or 180° about the flexion–extension axes. In both A and B, anatomical coordinate systems used to define joint coordinate systems are illustrated; blue (*z*-axis) corresponds to flexion–extension, green (*y*-axis) corresponds to abduction–adduction and red (*x*-axis) corresponds to long-axis rotation. (C, D) The 36 muscle–tendon actuators for the right leg, shown for an arbitrary standing posture in lateral (C) and anterior (D) views. (E–G) Close-ups on different parts of the model to show the actuator paths used to model muscle–tendon units, along with various wrapping surfaces (blue and purple geometries) used to help constrain these paths. (H) A single contact sphere was applied to the digits (terminal) segment of each limb, with radius and location as shown.

*Segment mass properties.* To derive the mass properties (mass, centre of mass [COM] location and inertial tensor) of each segment in the musculoskeletal model, the intact carcass of a separate tinamou individual (DDT07) was CT scanned (Toshiba Aquilion ONE, 120 kV peak tube voltage, 75 mAs exposure, 500 ms exposure time, 0.25 mm slice thickness, 0.427 mm pixel resolution) in a roughly 'neutral' and symmetric standing position with the wings folded and the neck and head directed cranially and straight out from the trunk. It was then disarticulated to produce the major segments, which were massed and then re-scanned (same machine settings as above, but with 0.407 mm pixel resolution) to digitally measure segment volume and in turn calculate segment-specific densities (reported in S1 Table). The geometries of flesh

and bones in the intact carcass CT scans were segmented using a combination of manual and automated techniques in Mimics, and the flesh segments were scaled and reoriented to the bones of the DDT09 Maya marionette via affine transformation, using the point registration tool in CloudCompare 2.5.4 (http://www.cloudcompare.org/). Two flesh geometries (trunk + wings, neck + head) were modelled for the 'trunk' body in the model, and flesh segments of the left hindlimb were mirrored for the right hindlimb, to maintain bilateral symmetry. The scaled flesh geometries were then used to compute segment mass properties as per [103,104], incorporating segment-specific densities, as part of the above process of transcribing the Maya model to the OpenSim environment. To ensure bilateral symmetry in the model, the mediolateral position of the trunk body COM was set to lie perfectly on the body midline.

**Muscle–tendon unit paths.** The paths of each muscle–tendon unit (MTU) were reconstructed in OpenSim with reference to anatomical dissection of DDT09, supplemented with observations made of other tinamou dissections and comparison to Suzuki et al. [105]. These paths were constrained to follow realistic lines of action using a combination of via points and cylindrical, ellipsoidal and toroidal wrapping surfaces (Fig 2E–2G, [106–108]), and only the minimum number of these required to achieve realistic paths were used for each muscle. To provide additional clarification, the 3-D geometries of muscles in the right hindlimb of a separate specimen (DDT03) were obtained through iodine staining [109] (Lugol's solution, 4% iodine with 10% neutral-buffered formalin, stained for 3.5 months with solution refreshed every two to three weeks) and micro-CT scanning (Nikon XTEK XTH 225 ST, 190 kV peak tube voltage, 245 mA tube current, 1,000 ms exposure time, 0.0907 mm isotropic voxel resolution), followed by manual and automated segmentation in Amira 2019.2 (ThermoFisher Scientific Inc., Waltham, USA). This separated the muscle bellies of almost all muscles that crossed the hip and knee joints. The geometries of these bellies were then imported into previously published MATLAB code [48] to determine lines of action, taken as the path passing through the centroid of 50 slices running transversely across the 3-D muscle volume. The bone geometries and muscle paths were linearly scaled to the size of DDT09 and imported into Maya, wherein the marionette of DDT09 was rotoscoped to match the *in situ* posture of the DDT03 cadaver. Once the bones were aligned between the two birds, the muscle paths of DDT03 were transferred to the bones of DDT09 and subsequently followed these bones through the process of OpenSim model generation. Thus, the muscle paths in their correct spatial relationships to the bones were directly visible in the OpenSim graphical interface, enabling more detailed refinement of MTU paths. Once the left limb's MTU paths were completely specified, the whole limb (bones, muscles, joints and mass properties) was mirrored about the sagittal plane to produce a perfectly symmetric model.

A total of 36 MTU actuators representing 34 separate muscles–all the important muscles of the hindlimb insofar as locomotion is concerned–were included in each limb of the musculoskeletal model (Fig 2C and 2D and Table 2; see also S1 Fig). As with Hudson et al. [110], but in contrast to Suzuki et al. [105], a caudofemoralis pars caudalis was not observed and so was not modelled. Either this muscle was genuinely lacking in the current study's specimen, or it was so small as to be indistinct from surrounding muscles (principally, the caudofemoralis pars pelvica); in the latter scenario its (very small) mass and force-generating capacity would have been incorporated into that of surrounding muscles. A similar situation applies to the fibularis brevis, which was not observed in the present study and so not modelled. On account of their broad area of origin on the ilium, the iliotibialis lateralis pars postacetabularis (ILPO) and iliotrochantericus caudalis (ITCa) were modelled with two MTUs. Two very small crural muscles, the popliteus and the plantaris, were not modeled here as their contributions to locomotion were deemed minimal enough to be safely ignored; moreover, the popliteus spans the tibiotarsus and fibula, which were immobile relative to one another in the model. The small intrinsic

**Table 2. Muscle–tendon units included in the musculoskeletal model.** The original architectural parameters as derived from anatomical dissection are listed (values for fibre length and pennation angle are means of multiple measurements), and tuned fibre and slack lengths are given in parentheses. The mass and maximal isometric force for the ILPO and ITC was split evenly between two actuators.

| | Abbreviation | Belly mass, $m$ (g) | Fibre length, $\ell_o$ (mm) | Tendon slack length, $L_S$ (mm) | Pennation angle, $\alpha_o$ (°) | Maximal isometric force, $F_{max}$ (N) |
|---|---|---|---|---|---|---|
| Iliotibialis cranialis | IC | 2.49 | 62.2 (70.3) | 3.1 (3.1) | 0 | 11.324 |
| Iliotibialis lateralis pars preactabularis | ILPR | 0.521 | 28.8 (31.2) | 39 (45.2) | 0 | 5.118 |
| Iliotibialis lateralis pars postacetabularis (anterior part) | ILPOa | 2.15 | 65.4 (69.9) | 2.7 (2.7) | 0 | 9.814 |
| Iliotibialis lateralis pars postacetabularis (posterior part) | ILPOp | 2.15 | 65.4 (749) | 1.3 (1.3) | 0 | 9.814 |
| Ambiens | AMB | 0.3 | 21.5 (23.1) | 24.6 (28.6) | 0 | 3.952 |
| Femorotibialis lateralis | FMTL | 1.565 | 28 (37.1) | 0.6 (0.6) | 21 | 14.913 |
| Femorotibialis intermedius | FMTI | 0.449 | 13.6 (14.7) | 25.4 (32.5) | 22.5 | 9.33 |
| Femorotibialis medialis | FMTM | 2.09 | 31.4 (31) | 16 (15.5) | 35.3 | 18.845 |
| Iliofibularis | ILFB | 2.133 | 70 (71) | 0.7 (0.7) | 0 | 8.263 |
| Flexor cruris lateralis pars pelvica | FCLP | 4.52 | 63.6 (77.4) | 4.1 (4.2) | 0 | 20.101 |
| Flexor cruris lateralis pars accessoria | FCLA | 0.844 | 32.8 (39.9) | 23.8 (30.4) | 0 | 7.276 |
| Flexor cruris medialis | FCM | 0.247 | 36.7 (41.3) | 15.3 (16.8) | 0 | 1.907 |
| Iliofemoralis externus | IFE | 0.212 | 9.8 (11.3) | 5.3 (6) | 0 | 6.098 |
| Iliotrochantericus cranialis | ITCr | 0.325 | 14.4 (15.8) | 11.4 (13.1) | 7.8 | 6.378 |
| Iliotrochantericus medius | ITM | 0.066 | 10 (12.9) | 2.9 (3.3) | 10.4 | 1.86 |
| Iliotrochantericus caudalis (anterior part) | ITCaa | 1.35 | 15.5 (19.1) | 12.1 (16.1) | 26.2 | 24.65 |
| Iliotrochantericus caudalis (posterior part) | ITCap | 1.35 | 15.5 (23.2) | 1 (1.1) | 26.2 | 24.65 |
| Ischiofemoralis | ISF | 0.56 | 14.3 (15.5) | 13.8 (15.8) | 24.8 | 11.114 |
| Obturatorius medialis | OM | 0.619 | 12 (13.6) | 20.2 (22.4) | 30 | 14.655 |
| Obturatorius lateralis | OL | 0.0320 | 14.7 (14.7) | 0.2 (0.2) | 0 | 0.624 |
| Caudofemoralis pars pelvica | CFP | 1.009 | 26.7 (32.8) | 0.2 (0.2) | 0 | 10.687 |
| Puboischiofemoralis medialis et lateralis | PIFML | 1.543 | 36 (38.5) | 0.3 (0.3) | 0 | 12.134 |
| Gastrocnemius pars lateralis | GL | 2.55 | 24.2 (26.6) | 59.2 (68.3) | 30 | 29.816 |
| Gastrocnemius pars intermedia | GI | 0.438 | 36.2 (39.3) | 44 (53.5) | 15 | 3.425 |
| Gastrocnemius pars medialis | GM | 2.251 | 25.2 (27.4) | 50.2 (64.2) | 20.8 | 25.305 |
| Fibularis longus | FL | 2.315 | 27.1 (28.2) | 105.6 (118.4) | 25.4 | 24.198 |
| Tibialis cranialis capute femorale | TCf | 0.519 | 15.6 (17) | 62.4 (67.3) | 24.8 | 9.415 |
| Tibialis cranialis caput tibiale | TCt | 0.979 | 30.6 (37.3) | 31.3 (38.9) | 18.5 | 9.049 |
| Extensor digitorum longus | EDL | 0.295 | 17.8 (17.9) | 109.7 (112.6) | 5 | 4.688 |
| Flexor digitorum longus | FDL | 0.563 | 26.7 (27.4) | 89 (104.7) | 25.6 | 5.963 |
| Flexor hallucis longus | FHL | 0.304 | 16.8 (16.9) | 135.6 (143.7) | 27 | 5.124 |
| Flexor perforatus digitorum II | FP2 | 0.524 | 20.4 (20.6) | 116.6 (124.4) | 22.4 | 7.277 |
| Flexor perforans et perforatus digitorum II | FPP2 | 0.369 | 19.4 (19.6) | 119.3 (130.6) | 30 | 5.378 |
| Flexor perforatus digitorum III | FP3 | 0.264 | 16.2 (16.3) | 128.2 (139) | 30 | 4.615 |
| Flexor perforans et perforatus digitorum III | FPP3 | 0.214 | 17 (17.1) | 135.3 (140.8) | 22 | 3.569 |
| Flexor perforatus digitorum IV | FP4 | 0.263 | 19.2 (19.4) | 115.6 (125.4) | 25 | 3.879 |

muscles of the pes (e.g., abductors, extensores proprii, lumbricales) were also ignored, largely due to the modelling simplification of treating all digits as a single segment. In a similar fashion to previous avian models [49,111], both parts of the flexor cruris lateralis, partes pelvica (FCLP) and accessoria (FCLA), were modelled, following the same line of action proximally towards the pelvis but diverging distally towards the tibiotarsus and femur, respectively.

**Muscle–tendon unit parameterization.** Muscle architectural data for DDT09 (Table 2) were measured during dissection following standard protocols [e.g., 111,112], and included muscle belly mass, resting fibre length and pennation angle ($\alpha_o$). The measurements obtained were found to be comparable to that measured in a second set of tinamous as part of a separate study (A.R. Cuff et al., in prep.). Architectural data were then used to estimate each muscle's maximal isometric force as

$$F_{max} = \frac{m \cdot \sigma}{\rho \cdot \ell_O}, \qquad (1)$$

where $m$ is muscle belly mass, $\sigma$ is the maximum stress developed in the fibres and $\rho$ is muscle tissue density; this assumes that the measured resting fibre length was equivalent to $\ell_o$ in subsequent use of a Hill-type model of contraction dynamics [5]. The values used for $\sigma$ and $\rho$ were taken as standard for vertebrate skeletal muscle: 300,000 N/m$^2$ [45,64,74,113] and 1060 kg/m$^3$ [111,114], respectively. Note that the calculated $F_{max}$ does not yet factor in the effect of $\alpha_o$, for this is taken into account in the geometric underpinnings of the Hill-type formulation used in the simulations outlined below; that is, $\alpha_o$ is explicitly accounted for in the maintenance of constant muscle thickness [5,7,50,115]. The values thus obtained for $F_{max}$, $\ell_o$ and $\alpha_o$ were assigned directly to the corresponding MTUs of the model. An exception was the ILPO and ITCa, since these were modelled with two MTUs; in both cases, mass and $F_{max}$ obtained for these muscles was distributed equally between their respective MTUs, but $\ell_o$ and $\alpha_o$ remained unaltered from the original measurements. In addition to architectural measurements of muscle bellies, the lengths and masses of the corresponding tendons were also measured, for estimation of tendon stiffness in the inverse and static simulations (below). Following assignment of architectural properties and definition of MTU paths, tendon slack length ($L_S$) for each MTU was estimated using the approach of Manal and Buchanan [116]. This parameter denotes the length at which the in-series tendon is slack (i.e., resting length), which in practice is difficult to determine empirically due to the complicating effect of internal tendon or aponeuroses. The calculation of tendon slack length was implemented assuming that muscle fibres had the capability to range in length from 0.5–1.5× $\ell_o$ across the limb's entire range of motion. Whilst the underlying model of muscle contraction used in the procedure of Manal and Buchanan [116] is different to that used in the inverse and static simulations, it should be noted that this was only used to obtain a first approximation of $L_S$, since this was subject to modification in the inverse simulations (below). Maximal fibre contraction velocity was set to 10× $\ell_o$/second.

## Synthesizing experimental data and the model

As the processed experimental kinematic and kinetic data were expressed in the same global coordinate system, and the kinematics was also expressed in the same joint coordinate system as the musculoskeletal model (albeit with fixed offsets applied to flexion–extension angles), the data were able to be fed directly into the musculoskeletal model in the OpenSim modelling environment. Moreover, as the two birds used for the focal trials were of near-identical size, the kinematics obtained for the running trial (DDT04) was fed directly into the model (based on DDT09) without being scaled. Nevertheless, due to nuances and limitations involved with the generation of the kinematic data, the lack of translational DOFs in the musculoskeletal model and the simplified representation of the 'trunk' segment, appreciable dynamic inconsistency existed between the kinematics, kinetics and the model (for the walking trial), as well as discrepancy between digit placement during the stance phase and the location of the centre of pressure of the GRF (walking trial) or the ground plane (both trials).

**Residual reduction algorithm.** The above dynamic inconsistency was rectified by first applying OpenSim's residual reduction algorithm to the walking trial data, using a two-pass process [101,117]. In the first pass, the algorithm was run with high tracking weights on the input kinematics and was allowed to modify the mass and COM location for the most massive segment in the body (trunk) to improve the match between kinematics and kinetics. This resulted in the trunk COM being shifted 5 mm caudally and 5 mm ventrally in the pelvis coordinate system (equivalent to ~6% of glenoacetabular distance), and the total body mass being increased by 2.1%, which was applied equally to all body segments; this scaling factor was also applied to the inertial tensor of each body segment as well, but COM location remained unaltered (effectively, density was increased, but density distribution remained the same). In the second pass, the adjusted model was then used with the original kinematic and kinetic data, with low tracking weights on the kinematics to allow the algorithm to modify the kinematics so as to reduce dynamic inconsistency (i.e., the magnitude of force and moment residuals at the ground–pelvis joint). Despite this procedure, the resulting residuals remained unacceptably high, sometimes exceeding double the threshold suggested by Hicks et al. [118] (particularly for moments), necessitating additional modification to the input data of the model.

**Tracking simulations.** To achieve an acceptable level of dynamic consistency between the model, kinematics and kinetics, a tracking simulation of walking was performed using a direct collocation approach [119,120]. Here, the rigid body dynamics component of the musculoskeletal model (with modified segment mass properties derived from the residual reduction step above) was transcribed to an algorithmically differentiable C++ source file, and called as an external function within a custom MATLAB script that formulated the tracking simulation as an optimal control problem (see S1 Code). The model was driven by torque actuators acting at each joint, and was constrained to move at the same average speed as in the trial, whilst tracking as closely as possible the recorded kinematics and GRFs:

$$
\min J = \int_{t_{\text{initial}}}^{t_{\text{final}}} \left( \begin{array}{l} w_1 \sum_{l=1}^{L} a_{\text{torque},\, l}^2 + \\[2ex] w_2 \sum_{n=1}^{N} (q_{n,\, \text{sim}} - q_{n,\, \text{exp}})^2 + \\[2ex] w_3 \sum_{k=1}^{K} (R_{k,\, \text{sim}} - R_{k,\, \text{exp}})^2 + \\[2ex] w_4 \sum_{n=1}^{N} \ddot{q}_n^2 \end{array} \right) dt,
\tag{2}
$$

for $L$ torque actuator activations $a_{\text{torque}}$ (= 20 in the current model), $N$ pairwise deviations between model coordinates in the simulation $q_{\text{sim}}$ and the kinematic data $q_{\text{exp}}$ (= 26 in the current model), $K$ pairwise deviations between each GRF component for the right and left feet in the simulation $R_{\text{sim}}$ and kinetic data $R_{\text{exp}}$ (= 6 in the current model), and $N$ model coordinate accelerations. Due to differences between the contact model used (involving GRMs in all three axes; see below) and experimental data (only the vertical component of the GRM is non-zero), the simulation was not encouraged to track the experimentally recorded GRMs. The values for activation and acceleration weighting terms were low ($w_1 = w_4 = 0.1$) whilst the weightings on the tracking terms were set to high values ($w_2 = 200$, $w_3 = 10$) to ensure that the solution maintained high fidelity to the original experimental data. The torque actuators were modelled with excitation–activation dynamics (basic first-order differential equation, approximating time delay) to produce a smoother solution for joint moments. To simulate GRFs and GRMs in the

problem, a single contact sphere was applied at the digits segment (Fig 2H), and foot–ground contact was modeled using a smoothed implementation of OpenSim's Hunt–Crossley formulation [119–121], with contact stiffness set to 250,000 N/m. It should be noted that both the GRFs and GRMs applied to the digits segments were derived strictly from the kinematics of the digits and their attached contact spheres. Unlike the residual reduction algorithm, the tracking simulation approach allows for both kinematics and kinetics to be altered in the course of the simulation. By enforcing Newtonian rigid body dynamics as a path constraint in the optimal solution, this guaranteed that at the converged solution, the output kinematics and kinetics were dynamically consistent with each other and the model (within a specified tolerance, here $10^{-4}$). The optimal control problem was discretized across 200 evenly spaced mesh intervals using CasADi 3.4.5 [122], and solved using the solver IPOPT 3.12.3 [123]. The resulting optimal solution involved generally only minor changes to the kinematics, GRFs and (vertical) GRM compared to the original experimental data (Table 3; see also S2 Fig). Greater modification occurred to the MTP angle, which is not surprising given the simplified representation of the distal pes (a single DOF, rather than 12 or more) and its contact with the ground (a single large contact sphere) in the model. It was deemed acceptable for the purposes of the current study that realism in the distal pes was sacrificed for greater kinematic and kinetic realism in the remainder of the limb. Modification to the GRFs was on average less than 10% of body weight, with peak deviations of 14.7%, 28.5% and 7.5% of body weight in the $x$, $y$ and $z$ directions, respectively. Deviation between the vertical GRM in the simulation and as measured experimentally (the 'free vertical moment') was on average less than 2% of the product of body weight and standing hip height, with a peak deviation of 7.8% of the product of body

**Table 3. Comparison of the results of the tracking simulations in relation to the original experimental data used as input.** This is reported as root mean squared error for kinematics and kinetics; error is reported in degrees for rotational kinematics, millimetres for translational kinematics, Newtons for GRFs and Newton-metres for GRMs. Note that only the vertical component of the GRM is considered, as horizontal components in reality are practically zero (yet non-zero components can exist in the simulation, rendering any comparison approximate only). For entries pertaining to the limbs, error is reported as left limb/right limb.

| Data | Walking | Running |
|---|---|---|
| Pelvis yaw | 1.512 | 3.612 |
| Pelvis pitch | 2.301 | 4.519 |
| Pelvis roll | 0.983 | 4.493 |
| Pelvis Tx | 5.415 | 5.459 |
| Pelvis Ty | 1.776 | 3.218 |
| Pelvis Tz | 8.195 | 2.431 |
| Hip extension | 1.123/1.177 | 0.956/0.976 |
| Hip abduction | 1.156/1.237 | 0.340/0.418 |
| Hip rotation | 1.831/1.942 | 0.702/0.569 |
| Knee extension | 4.250/3.295 | 4.165/3.192 |
| Knee abduction | 1.736/1.588 | 0.408/0.436 |
| Knee rotation | 3.073/3.300 | 1.104/0.979 |
| Ankle extension | 3.592/4.347 | 3.602/3.891 |
| Ankle abduction | 1.224/1.465 | 1.089/0.833 |
| Ankle rotation | 4.009/4.077 | 0.885/0.397 |
| MTP angle | 12.986/8.850 | 19.437/31.943 |
| GRF$_x$ | 0.168/0.140 | — |
| GRF$_y$ | 0.496/0.258 | — |
| GRF$_z$ | 0.117/0.181 | — |
| GRM$_y$ | 0.011/0.03 | — |

weight and standing hip height. It should be recognized that in the present simulation framework there exists an interdependency of GRFs, GRMs and kinematics; for example, the centre of pressure of the GRF (which in turn affects the computed GRMs as expressed in a local or global reference system) is dictated by the kinematics of the body segment to which the contact sphere is appended, and displacement of the centre of pressure will therefore affect the achievement of force and moment balance (i.e., dynamic consistency).

A tracking simulation was also performed for the running trial, so as to generate ground reactions that were dynamically consistent with the experimentally derived kinematics, and to improve digit placement with respect to the ground plane during the stance phase. The same torque-driven optimal control problem formulation used for the walking simulation was also used here. However, in lieu of experimentally derived kinetic data, the inputs used for the GRF in the $x$ (anteroposterior) and $y$ (vertical) directions were estimated using the BIRDS model of Bishop et al. [28], given the known values for speed and duty factor. The $x$ and $y$ components of the simulated GRFs were then encouraged to track these estimates in the simulation; the $z$ (mediolateral) component was allowed to freely vary so as to allow dynamic consistency. Given the highly simplified representation of foot–ground interaction in the current model, tracking just the (less than ideal) kinematic data alone would have resulted in unrealistic GRF profiles. Forcing the simulation to track a theoretically expected GRF (which is based on prior empirical observations) in addition to the kinematic data, kept the simulation better grounded in biological realism. One further difference from the walking simulation was that a lower contact stiffness (150,000 N/m) was found to produce markedly better results in terms of tracking error; this is consistent with the empirical observation of greater limb compliance at faster speeds in a variety of species [124,125]. As with the walking trial, the resulting optimal solution involved generally minor changes to the kinematics compared to the original data, again except for MTP angle (Table 3). In addition to producing modified locomotor kinematics (and kinetics), the above procedure also computed the external joint moments as would be calculated via an inverse dynamics routine (i.e., the moments incurred about each joint due to gravity and inertia). These formed the final inputs used in the inverse simulation below.

## Inverse simulation

The 'load sharing' or 'inverse' problem of deriving muscle recruitment patterns that produced the observed (or more strictly speaking, XROMM-inspired) kinematic patterns for the walking and running trials was addressed using MATLAB code in the Muscle Redundancy Solver package (v 2.1; https://simtk.org/projects/optcntrlmuscle) of De Groote et al. [7]. Briefly, this package solves for muscle excitation time histories that produce the external joint moments while accounting for muscle dynamics. The resulting dynamic optimization is implemented by formulating the optimal control problem as a nonlinear program via direct collocation. MTUs are modelled with a Hill-type model that uses implicit, algorithmically differentiable representations of excitation–activation and activation–contraction dynamics, with activation and tendon force as state variables. In the current study, tendon force was used as a state variable with normalized fibre length ($\ell^* = \ell/\ell_\mathrm{o}$) as an output, the latter or which is often a state variable in other implementations of the Hill-type model [e.g., 6].

**Muscle redundancy solver modifications.** In the approach of De Groote et al. [7], the force–length curve of the in-series tendon is in part parameterized by dimensionless tendon stiffness, calculated as

$$k_\mathrm{T} = \frac{E \cdot A_\mathrm{T}}{F_\mathrm{max}} = \frac{E \cdot A_\mathrm{T} \cdot \rho \cdot \ell_\mathrm{O}}{m \cdot \sigma \cdot \cos(\alpha_\mathrm{O})}, \tag{3}$$

where $E$ is Young's modulus for tendon, $A_T$ is the cross-sectional area of the tendon and $F_{max}$ is the maximal isometric force for the muscle. Note that in this calculation $F_{max}$ takes muscle pennation into account, in contrast to Eq 1 above, because the muscle is here reduced down to a force along a line of action in series with the tendon (i.e., fibre kinematics are irrelevant). In the human model of De Groote et al. [7], $k_T$ was nominally set to 35 [cf. 5]. However, anatomical measurements of all MTUs for the tinamou, including tendon length and mass, indicated that $k_T$ was in general markedly higher, assuming typical values of 1.2 GPa and 1,120 kg/m$^3$ for tendon modulus and density, respectively [126]. Values for $k_T$ ranged from 23 for some digital flexors to well over 1,000 for some hip muscles with short, thick tendons, with a median value of 114. As such, a first-pass estimate of $k_T = 100$ was used for all MTUs in the current study. The markedly higher stiffness inferred for many of the tendons in the tinamou, in comparison to humans, is consistent with prior empirical studies showing that smaller animals tend to have lower ratios of muscle physiological cross-sectional area compared to tendon cross-sectional area [e.g., 127,128–130]. Activation and deactivation time constants (in the muscle excitation–activation dynamics) were also altered from the default human model values of 0.015s and 0.06 s, to 0.007s and 0.027s, respectively. Although no published data exist to support these specific values, it was considered likely that both constants would scale with body size; smaller animals need to have less electromechanical delay due to being adapted for higher stride frequencies, and have a shorter distance for electrical signal to travel from motor neuron to muscle fibres [131,132]. Here it was assumed that the time constants would scale proportional to body mass$^{1/6}$ [29], and were scaled from the 64.8 kg human model to the 0.545 kg tinamou model in accordance with this. All other parameters regarding muscle excitation–activation and activation–contraction dynamics used the default values as per De Groote et al. [7].

The original Muscle Redundancy Solver code was further modified to solve for both walking and running trials together, whilst simultaneously also allowing MTU architectural parameters to be modified from their original values. The measurement of muscle architectural parameters during dissection inevitably involves a level of error, which may be magnified by simplified representation of an entire muscle effectively as a single fibre in Hill-type models [5]. It was therefore considered appropriate to allow some parameters to be adjusted from their original values such that the MTUs were better 'tuned' at executing the observed behaviours [22,133–136]. This helps reduce the discrepancy between model capabilities and measured performance, reducing reliance on reserve actuators (see also below), and also produces more physiologically plausible MTU behaviour. Here, $\ell_o$ and $L_S$ were tuned, with a single new set of parameters being derived that worked well for both walking and running trials. $F_{max}$ was kept at its original value for all MTUs; for a given physiological cross-sectional area, this implicitly assumes that the number of fibres varies in inverse proportion to $\ell_o$. By tuning for different gaits simultaneously, this reduced the chance of the MTUs becoming too well adapted for the execution of any single behaviour and possibly less effective in the execution of other behaviours. As in real life, an MTU's parameters will reflect a tradeoff in requirements for the effective execution of different behaviours throughout daily activity. Ideally, experimental data from a wider range of behaviours could be used to obtain even more 'generically adapted' MTU parameters, such as maximal speed running, sit-to-stand or jumping manoeuvres; unfortunately, suitable data were unavailable for the present study.

**Problem formulation and solving.** From the supplied joint kinematics, MTU length ($l_{MT}$) and moment arms throughout the trials were computed for each MTU using the MuscleAnalysis tool in OpenSim. The resulting data, and the external joint moments, were first filtered using a fourth-order low-pass Butterworth filter with a cutoff frequency of 20 Hz, prior to being used in the inverse simulations. This second pass of filtering was applied to remove additional noise or artifact that had collectively accumulated throughout the data

processing steps, and kept the focus of the simulations and analyses on the major patterns in the data (especially considering only a single trial each of walking and running were studied here). As tinamou movement was quite bilaterally symmetrical, MTU excitations were only solved for the left limb, which encompassed complete and contiguous stance and swing phases in both trials. In addition to the 36 MTUs, all ten DOFs in the left hindlimb were actuated by torque or 'reserve' actuators. These reserve actuators account for modelling simplifications or errors that collectively make it difficult, if not impossible, for the MTUs alone to completely balance external joint moments [17,118]. The maximal moment-generating capacity of these actuators was generally set to be a low value (relative to peak external joint moment), to encourage the use of MTUs in countering joint moments and driving movement. However, for abduction–adduction and long-axis rotation at both the knee and ankle, these maximal capacities were set to higher values to achieve more realistic results (see Results); this is consistent with the muscles that actuate these DOFs being positioned so as to act largely in the flexion–extension plane of the joints. Given that previous *in vivo* kinematic studies have demonstrated considerable mobility of the avian knee and ankle in abduction–adduction or long-axis rotation [91,137], it was felt that including these additional DOFs in the simulations could be more insightful than excluding them *a priori*.

The optimal control problem was posed thus:

$$
\min J = \int_{t_{\text{initial}}}^{t_{\text{final}}} \left( \begin{array}{l} w_1 \left( \sum_{m=1}^{M} a^2_{\text{muscle, } m, \text{ walk}} + \sum_{m=1}^{M} a^2_{\text{muscle, } m, \text{ run}} \right) + \\[2ex] w_2 \left( \sum_{l=1}^{L} a^2_{\text{reserve, } l, \text{ walk}} + \sum_{l=1}^{L} a^2_{\text{reserve, } l, \text{ run}} \right) + \\[2ex] w_3 \left( \sum_{m=1}^{M} \frac{da^2}{dt}_{m,\text{walk}} + \sum_{m=1}^{M} \frac{da^2}{dt}_{m,\text{run}} + \sum_{m=1}^{M} \frac{dF_{\text{T}}}{dt}^2_{m,\text{walk}} + \sum_{m=1}^{M} \frac{dF_{\text{T}}}{dt}^2_{m,\text{run}} \right) + \\[2ex] w_4 \left( \sum_{m=1}^{M} F^2_{m,\text{passive,walk}} + \sum_{m=1}^{M} F^2_{m,\text{passive,run}} \right) \end{array} \right) dt +
$$

$$
w_5 \sum_{m=1}^{M} (p_\ell - 1)^2 + w_6 \sum_{m=1}^{M} (p_L - 1)^2, \tag{4}
$$

subject to muscle excitation–activation and activation–contraction dynamics, muscle–tendon equilibrium within each MTU, state continuity across successive mesh intervals throughout each trial (i.e., collocation defect constraints are zero), and net moment balance between MTU forces, reserve actuators and external joint moments (see S1 Code). The problem was discretized across 100 evenly spaced mesh intervals (using third order Radau collocation) for both trials in CasADi and solved using IPOPT, and the initial guess supplied for the state variables was seeded with the results obtained from a static optimization solution. Note that the Muscle Redundancy Solver simply solves for muscle states and controls given provided input experimental data, without imposing constraints on the initial or final conditions of the system; thus, non-cyclical experimental data will result in non-cyclical muscle results.

The objective function aimed to minimize six terms:

1. The sum of squared muscle activations ($a_{\text{muscle}}$) across all $M$ muscles for both trials, integrated across the trial; activations could range between 0 ('fully off') and 1 ('fully on'). The corresponding weighting used for this was $w_1 = 2$.

2. The sum of squared reserve activations ($a_{\text{reserve}}$) across all $L$ DOFs in the limb for both trials, integrated across the trials. In a similar fashion to the muscles, reserve activations could range between -1 and 1 (allowing torques of either sense to be produced), with the absolute magnitude of supplied torque computed by scaling activations by pre-defined maximum external joint moments. This term was weighted heavily relative to the previous term ($w_2 = 500$), penalizing the use of reserve actuators unless absolutely necessary.

3. The sum of squared time derivatives of muscle activations ($da/dt$) and MTU forces ($dF_T/dt$) across all $M$ muscles for both trials, integrated across the trials. The inclusion of this term is to improve numerical conditioning of the nonlinear program, avoiding situations for which these implicit controls are not uniquely defined by optimality conditions [120]. The corresponding weighting used for this term was $w_3 = 1$; it was found that lower values (as used in the human simulations of [7]) resulted in qualitatively similar excitation patterns, but which were markedly less smooth over time.

4. The sum of squared passive muscle fibre forces ($F_{\text{passive}}$) across all $M$ muscles for both trials, integrated across the trials. The corresponding weighting used for this term was $w_4 = 100$. A high value here helps penalize the optimizer from 'cheating' and tuning $\ell_{\text{o}}$ such that a given MTU operates largely or wholly on the descending limb of its active force–length curve, thereby using passive forces to help counter joint moments with no associated activation cost [138]. Whilst this term may artefactually decrease or constrain the recovered operating ranges of some muscle fibres, it was found that preliminary simulations that did not include this term resulted in a few muscles optimizing their operating ranges at very high (and unrealistic) values of $\ell^*$. The weighting value ultimately used was chosen as a compromise that avoided these unrealistic results but minimized potential side-effects on other muscles.

5. The sum of squared deviations in $\ell_{\text{o}}$ tuning factors ($p_\ell$) from one; encouraging the tuned fibre lengths to differ only as much as is needed from the original values in the musculoskeletal model. The corresponding weighting used for this was $w_5 = 2$.

6. The sum of squared deviations in $L_S$ tuning factors ($p_L$) from one; encouraging the tuned slack lengths to differ only as much as is needed from the original values in the musculoskeletal model. The corresponding weighting used for this was $w_6 = 1$; a lower weighting is used compare to that for fibre lengths as greater confidence was placed *a priori* in the original values for fibre length, since these were directly measured whereas slack lengths were estimated. It was found that using higher values of $w_5$ in relation to $w_6$ did not have any important effect on the resulting excitation patterns.

Although the specific (absolute) values of the weighting terms were somewhat arbitrarily chosen (for both Eqs 2 and 4), their relative magnitudes reflected the relative importance of the corresponding terms in each objective function as per the study's objectives. A systematic exploration of the consequences of using differing weighting terms was beyond the scope of the present investigation, but could be useful in future comparative studies of larger datasets, especially across species.

## Static simulations

**Premise.** Following the execution of inverse simulations, the tuned musculoskeletal model was subject to four sets of static simulations. Only a subset of all possible limb postures was used *in vivo* by the tinamous in the locomotor experiments, and so these additional simulations helped explore the full range of possible fibre length variation across the entirety of

limb poses. Moreover, in concert with the results of the inverse simulations, a key aim of the static simulations was to explicitly test the assumption outlined in the Introduction, that $\ell^*$ will vary from 0.5–1.5 across the entire range of limb motion. Deconstructing this assumption, it was hypothesized that the length change an MTU experiences over the entire range of limb motion is equal to $\ell_o$ [66,73,74]; this in turn implies that $\ell^*$ will vary from approximately 0.5–1.5. Additionally, it was posited that by assuming a fixed moment arm, $l_{MT}$ change could therefore be computed directly from total joint range of motion. By exploring the full range of limb poses in the tinamou model, the static simulations therefore sought to elicit the greatest range of physiologically possible muscle lengths, providing a more comprehensive test of the aforementioned assumption than could be achieved with the *in vivo* locomotor simulations alone.

**Implementation.** A total of 5,000 random limb postures were generated across the model's full range of motion. For the sake of simplicity and tractable analysis of the results, the knee and ankle joint were reduced to a single DOF each, with abduction–adduction and long-axis rotation fixed at constant angles, equal to the mean angles across the stride for both the walking and running trials' kinematics. This simplification also obviated the need for reserve actuators, which were strongly required in the inverse simulations (see also Results), keeping the focus directed towards those DOFs for which muscles are primarily responsible for actuation (and therefore those DOFs most relevant to the assumption being tested). Thus, the 5,000 random limb postures resulted in a 6 DOF joint space. In defining range of motion limits, ranges for the knee ([-145˚, -10˚]), ankle ([-135˚, -0˚]) and MTP joint ([0˚, 180˚]) were set based on what is physiologically realistic, such as the knee not being permitted full extension due to soft tissue restrictions observed in intact and defleshed carcasses. For the 3-DOF hip, flexion–extension was nominally allowed full flexion through to full vertical extension (i.e., [-90˚, 0˚]), which is more extensive than that possible *in vivo*, as observed in cadaveric manipulations due to the limiting effects of intrinsic soft tissues of the hip [139], as well as far-field influences such as the ribcage. Ranges for hip abduction–adduction ([-5˚, 30˚]) and long-axis rotation ([-15˚, 30˚]) were determined from bone-on-bone collisions with the other two DOFs set at 0˚. This simplified approach did not accurately or completely capture the true nature of joint mobility in the hip (for which an automated method like that of [139] would be required), but it was deemed acceptable here for it almost certainly overestimated the total mobility of the joint [93]. It therefore enabled a conservative assessment of fibre length changes: the total scope of possible fibre length changes in reality would be less than what would be achieved here. Values for $l_{MT}$ and moment arms were then determined for the 5,000 test postures using the MuscleAnalysis tool in OpenSim.

For each test posture, four basic, static simulations were performed. In the first two, all 36 MTUs in the limb were considered together and an optimization problem was formulated, which sought to either minimize (simulation 1) or maximize (simulation 2) muscle activations,

$$\min/\max \sum_{m=1}^{M} a^2_{\text{muscle, } m} \, , \tag{5}$$

subject to maintaining static equilibrium throughout the leg,

$$\sum_{m=1}^{M} F_{m,k} \cdot r_{m,k} = 0 \text{ for all } k \text{ DOFs}, \tag{6}$$

and maintaining muscle–tendon equilibrium within each MTU; $r_{m,k}$ is the moment arm of muscle $m$ about DOF $k$. Gravity was ignored here as it is otherwise very difficult to standardize the results against limb posture to enable meaningful comparison; and as the simulations were

static, there were no external joint moments due to segment inertia. That is, only the geometric, skeletal aspect of the system was considered here. In the third and fourth simulations each MTU was considered sequentially in isolation, whereby an optimization problem was formulated in which activation was set to zero (simulation 3) or one (simulation 4) and then the MTU was solved for muscle–tendon equilibrium.

In all four simulations, MTU behaviour was modelled using the same formulation as in the inverse simulations, and the optimization problem in each was formulated in CasADi and solved with IPOPT. Simulations 2–4 arguably did not follow any physiologically relevant principle, but were included so as to circumscribe as great a range as possible of muscle shortening and lengthening, beyond what occurred *in vivo* in the locomotor experiments. For example, in considering a single muscle at a time, simulations 3 or 4 may elicit a greater range of fibre lengths than simulations 1 or 2 alone. Regardless, the ranges of $\ell^*$ possible in the real animal will be less than what was achieved here, and therefore the simulations lent themselves towards a conservative testing of the above assumption. In combination with the inverse simulations of *in vivo* locomotor behaviour, the static simulations may be seen as progressively simplified analyses of the tinamou hindlimb musculoskeletal system, focusing increasingly more on the muscles themselves (insofar as they relate to skeletal anatomy) in isolation of other complicating factors, namely dynamics, gravitational and inertial loading and muscle co-contraction.

**Results filtering and analysis.** Prior to analysis and interpretation, the results of the above simulations first required filtering, to remove poses that produced justifiably 'bad' results. Three such instances of this occurred:

1. Poses in which it was impossible to achieve muscle–tendon equilibrium within all MTUs and simultaneously achieve moment balance across all the joints, regardless of activation. The underlying cause of this was very high $\ell^*$ for at least one MTU, producing extremely high passive fibre forces. Only simulations 1 and 2 were affected by this.

2. Poses where $l_{MT} < L_S$ for a given MTU, in which case muscle–tendon equilibrium cannot be achieved regardless of activation and no solution therefore exists. Only simulations 3 and 4 were affected by this.

3. Poses where unrealistically high values of $l_{MT}$ were achieved for the flexores cruris lateralis (FLCA and FLCP) due to the poor wrapping behaviour for these muscles' actuator paths at extreme hip flexion (coupled with low hip abduction). This poor behaviour occurred in poses well exceeding those observed *in vivo* during locomotion or *in situ* during cadaveric manipulations (due to soft tissue limits on hip mobility); it was hence a consequence of the model performing poorly in a non-physiological situation that it was never intended to represent, due to the use of highly inclusive range of motion limits defined above. These instances could be easily identified from a histogram of $l_{MT}$ for the FCLA and by visual inspection of the musculoskeletal model in OpenSim; it was *a priori* set that poses with $l_{MT}$ of the FCLA exceeding 80 mm were considered inviable. This affected all four simulations.

Results for the viable poses were then examined with respect to the above assumption, via three comparisons. Firstly, the gross range of $\ell^*$ and $l_{MT}$ achieved for each MTU across the simulations was computed, to test if this met the expectation of 0.5–1.5$\times \ell_o$. Secondly, the range of $\ell^*$ and $l_{MT}$ achieved was compared muscle-by-muscle. Thirdly, as it was posited that $l_{MT}$ change could be computed from total joint range of motion by assuming a fixed moment arm, this implies that $\ell^*$ should vary approximately in proportion to (linearly with) joint angle (s). As such, a hyperplane was fitted to plots of $\ell^*$ and joint angle(s) using linear algebra:

$$\mathbf{C} = \mathbf{X}^+ \mathbf{y}, \tag{7}$$

where $\mathbf{C}$ is a $\eta \times 1$ vector of coefficients describing the hyperplane (of the form $c_1x_1 + c_2x_2 + \ldots + c_\eta$), $\mathbf{X}^+$ is the $\eta \times o$ Moore-Penrose pseudoinverse of an augmented matrix formed from the joint angle data and a column of ones, and $\mathbf{y}$ is a $o \times 1$ vector of the $\ell^*$ data, for $o$ observations over $\eta - 1$ joint angles. The resulting hyperplane is the minimum norm least-squares fit for the data; for comparisons involving a single joint angle, $\eta = 2$ and the hyperplane is a line, and for comparisons involving two joint angles, $\eta = 3$ and the hyperplane is a regular plane. Following the derivation of the hyperplane, the coefficient of determination ($r^2$), root mean squared error (RMSE) and maximum absolute error ($\varepsilon_{\max}$) were computed for the fit *versus* the original data. For each MTU, hyperplane fitting was applied twice: considering the results from simulations 1 and 2 together (analyses of all muscles holistically), and considering the results from simulations 3 and 4 together (analyses of each muscle individually). Hyperplane fitting, and the goodness of the resulting fit, therefore helped assess the scale of the effect of variable moment arms, compliant tendons and pennation angles on an otherwise linear relation between $\ell^*$ and joint angle(s).

## Results

### Inverse simulations

**Tuning factors.** Almost all tuning factors in the optimal solution exceeded one (Fig 3), indicating that most MTUs needed both $\ell_o$ and $L_S$ to be increased from their original values (Table 2). Whereas $\ell_o$ was increased on average by 11% (range -1.1 to 49.3%), $L_S$ was increased on average by 10.5% (range -3.1 to 33.3%). Therefore, many MTUs only required a level of tuning well within the scope of variation due to either measurement error (considering that a whole muscle is essentially represented as a single fibre) or genuine intra- or inter-individual variation. Moreover, the two most drastically altered MTUs can probably be explained in large part as a consequence of the simplified approach used to represent their paths in the model. Firstly, the FMTL ($\ell_o$ tuned by 32.4%, or 9.1 mm) crosses the knee joint over the lateral condyle region, and its involvement with the common aponeurosis of the knee extensors ('patellar tendon') is complex. It would not be surprising if the MTU path as modelled here deviates significantly from the 'true' path *in vivo* over at least part of the knee's range of motion, introducing error into muscle kinematics during movement, and thereby necessitating greater tuning in the simulation. Secondly, the posterior part of the ITCa (ITCap; $\ell_o$ tuned by 49.3%, or 9.1 mm) is one of two MTUs that was used to model a highly pennate muscle, and it is therefore possible that considerable tuning occurred here to compensate for this modelling simplification. It is worth noting that the anterior part (ITCaa) also had a high tuning factor for $\ell_o$ (+- 23.0%), and had the highest tuning factor for $L_S$ (+33.3%).

**Reserve contributions.** For both walking and running trials, reserve actuator contributions almost always remained less (often much less) than 10% of the external joint moment at any given instant in the stance phase, as well as during much of the swing phase, for the following DOFS: hip extension, abduction and rotation, knee extension, ankle extension and MTP angle (Fig 4). This indicated that the muscles provided the vast majority of the effort required to stabilize and propel the limb. Given that the original (dissection-derived) values for muscle $F_{\max}$ were used in these simulations, this result is encouraging, as even in the running trial the muscles were acceptably sufficient in sustaining the prescribed locomotor dynamics. The reserve contributions for the abduction–adduction and long-axis rotation DOFs of the knee and ankle were almost always markedly higher than 10% of the external joint moment, which at least in part stems from assigning a higher maximal moment-generating capacity to these actuators. In preliminary runs of the inverse simulations, it was found that whilst using lower maximal capacities led to lower reserve contributions to external joint moment, it also resulted

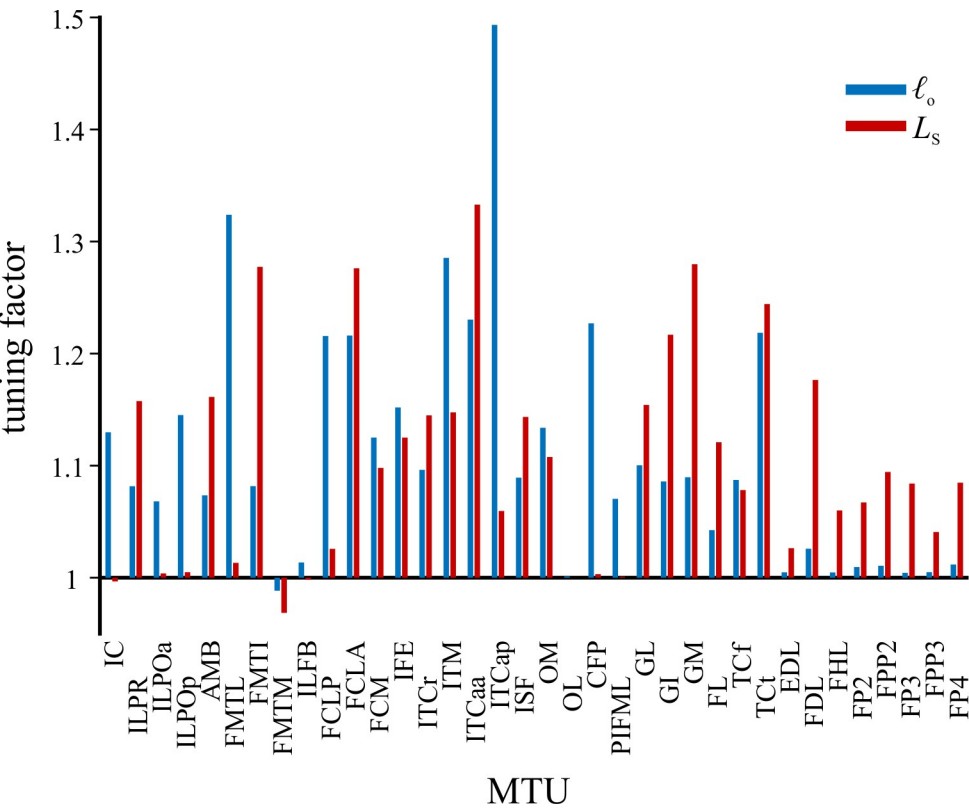

**Fig 3. Tuning factors used in the optimal solution for the inverse simulation.** These factors are multiplied against the original assigned values for optimal fibre length ($\ell_o$) and tendon slack length ($L_S$) to derive the tuned parameters for each MTU.

in higher excitations for muscles crossing the DOFs involved. The excitation patterns obtained for several of these muscles (particularly the ankle flexors and extensors) were grossly inconsistent with previously published empirical data as well as *a priori* expectations based on anatomy, and were therefore considered implausible. These results suggest an apparent necessity for increased use of non-muscular (i.e., passive) forces in controlling these particular DOFs (see below for further discussion).

**Muscle recruitment.** The time histories of muscle excitations recovered for the walking trial are presented in Fig 5, and those for the running trial in Fig 6. Excitation patterns are also visualized in the context of kinematics and kinetics in S1 and S2 Movies. There were no pronounced changes in muscle recruitment (timing or duration with respect to the stance and swing phases) between walking and running, except for generally greater excitations in running, corresponding to greater force production in this more strenuous gait. Exemplar comparisons of simulated excitations with previously published electromyographic data for walking and running birds are presented in Fig 7; as previous studies rarely quantified the magnitude of the recorded electrical signal (either in absolute terms, or in relation to a maximum voluntary contraction), comparisons with the literature are necessarily qualitative only. The recovered patterns of recruitment for many muscles were grossly similar to that recorded experimentally in birds [9,27,33,41,55,58,88,96,140], including for tinamous at similar speeds to those simulated here [88].

Muscles recruited exclusively or mostly during the stance phase included the ILPO, FMTL, FMTI, FMTM, FCLP, FLCA, ITCa, CFP, PIFML, GM, FL and the digital flexors. Muscles

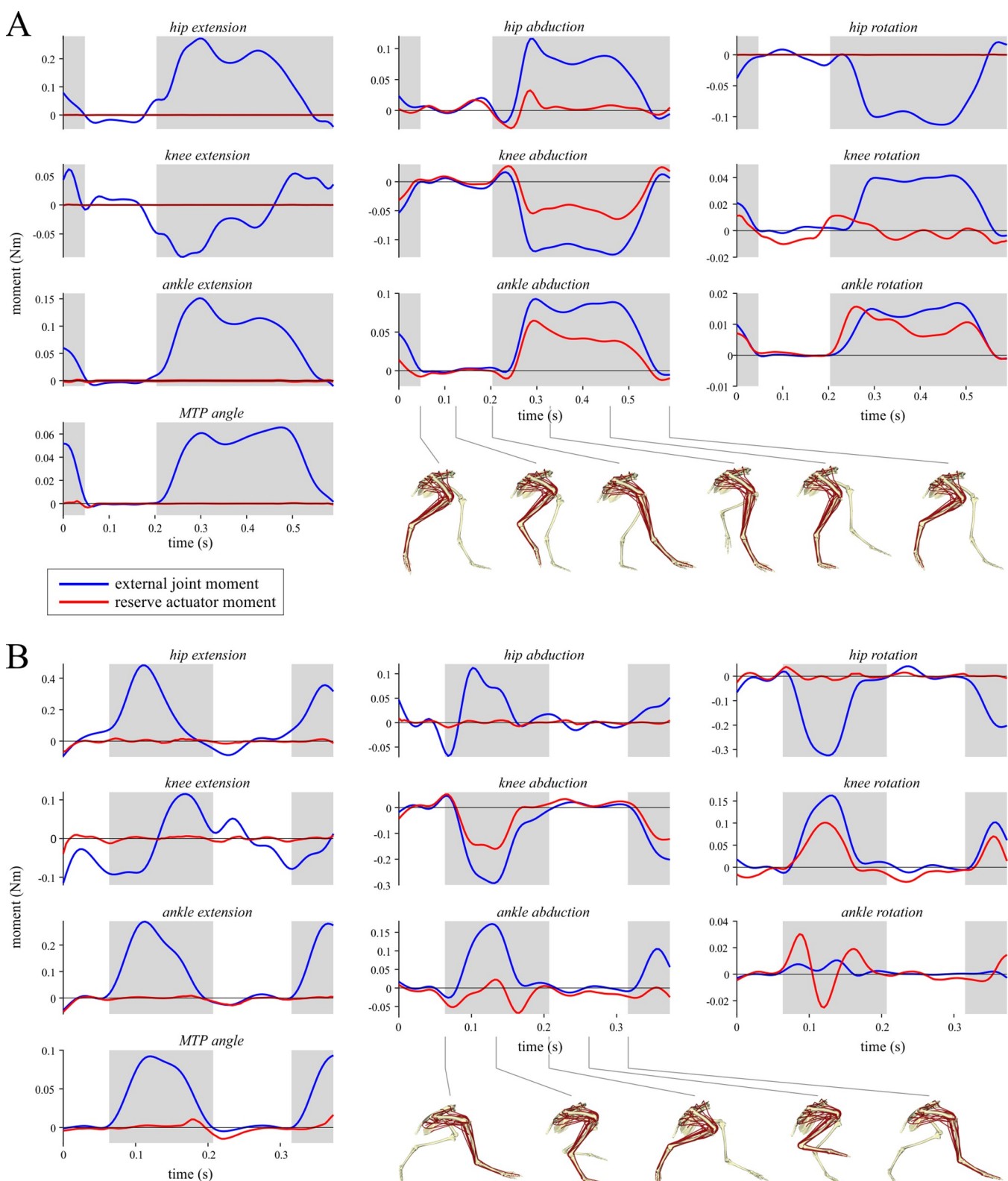

**Fig 4. Reserve actuator contributions to external joint moments.** This is shown for walking (A) and running (B) trials. Grey regions denote the stance phase, white regions denote the swing phase. Note the strong contributions to external joint moment made by reserves acting about the abduction–adduction and long-axis rotation DOFs of the knee and ankle joints. Also illustrated are the kinematics of the left hindlimb (reversed for visualization) at various parts of the stride cycle.

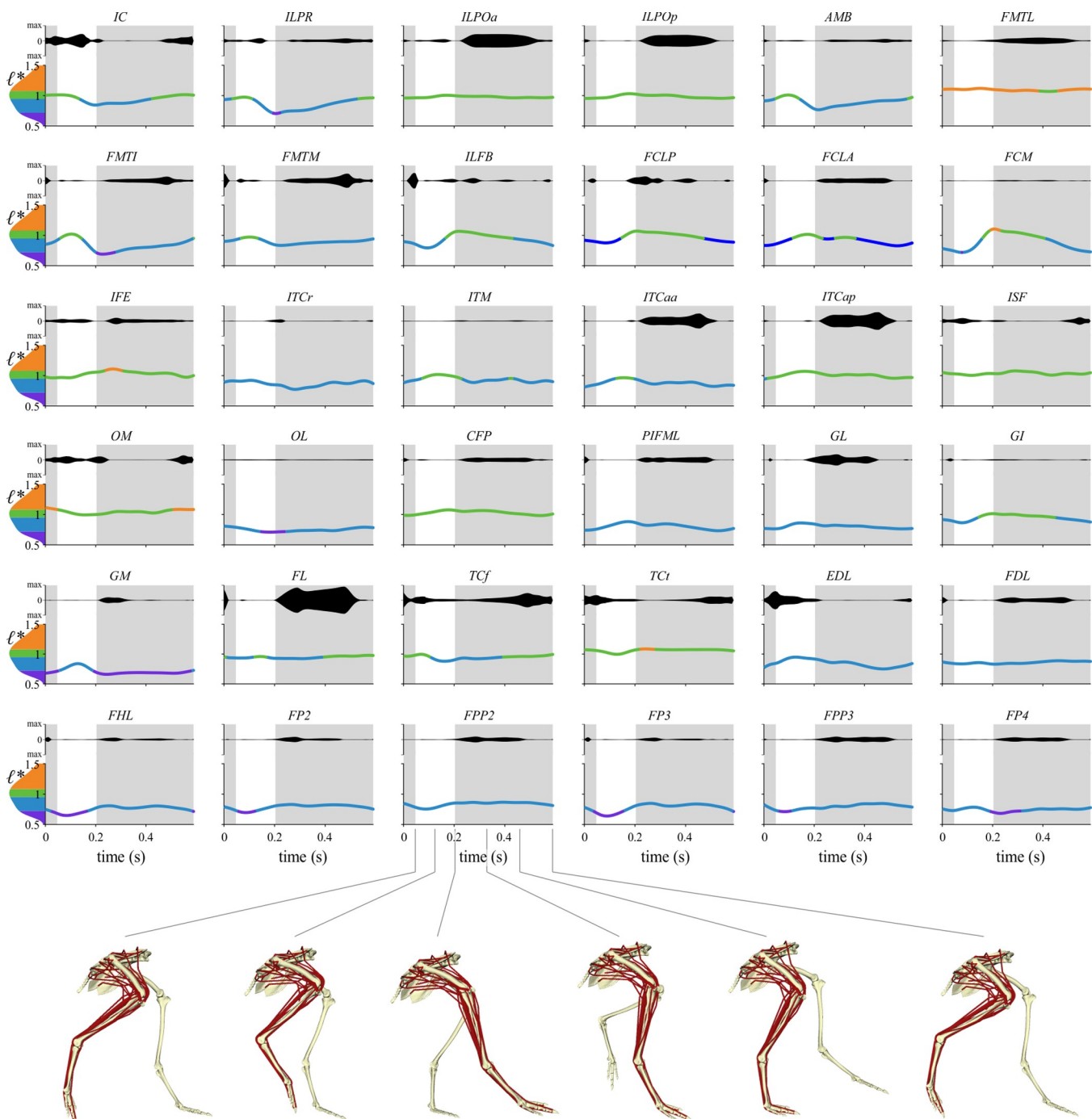

**Fig 5. Time histories of muscle excitation and fibre length change for each MTU in the walking simulation.** Excitations (from zero recruitment to possible maximal recruitment) are plotted above normalized fibre lengths for each MTU, the latter of which are colour coded according to where on the active force–length curve fibres are operating: purple = steep ascending limb, blue = shallow ascending limb, green = plateau, orange = descending limb (divisions approximately correspond to those of Arnold and Delp [15]). Excitation profiles obtained in the simulation are reflected about the abscissa so as to visually emulate the appearance of experimental EMG signals. Grey regions denote the stance phase, white regions denote the swing phase. See Table 2 for muscle abbreviations.

recruited exclusively or mostly during the swing phase included the IC, ILFB, OM and EDL. Several muscles showed small bouts of excitation during both swing and stance phases, the timing of

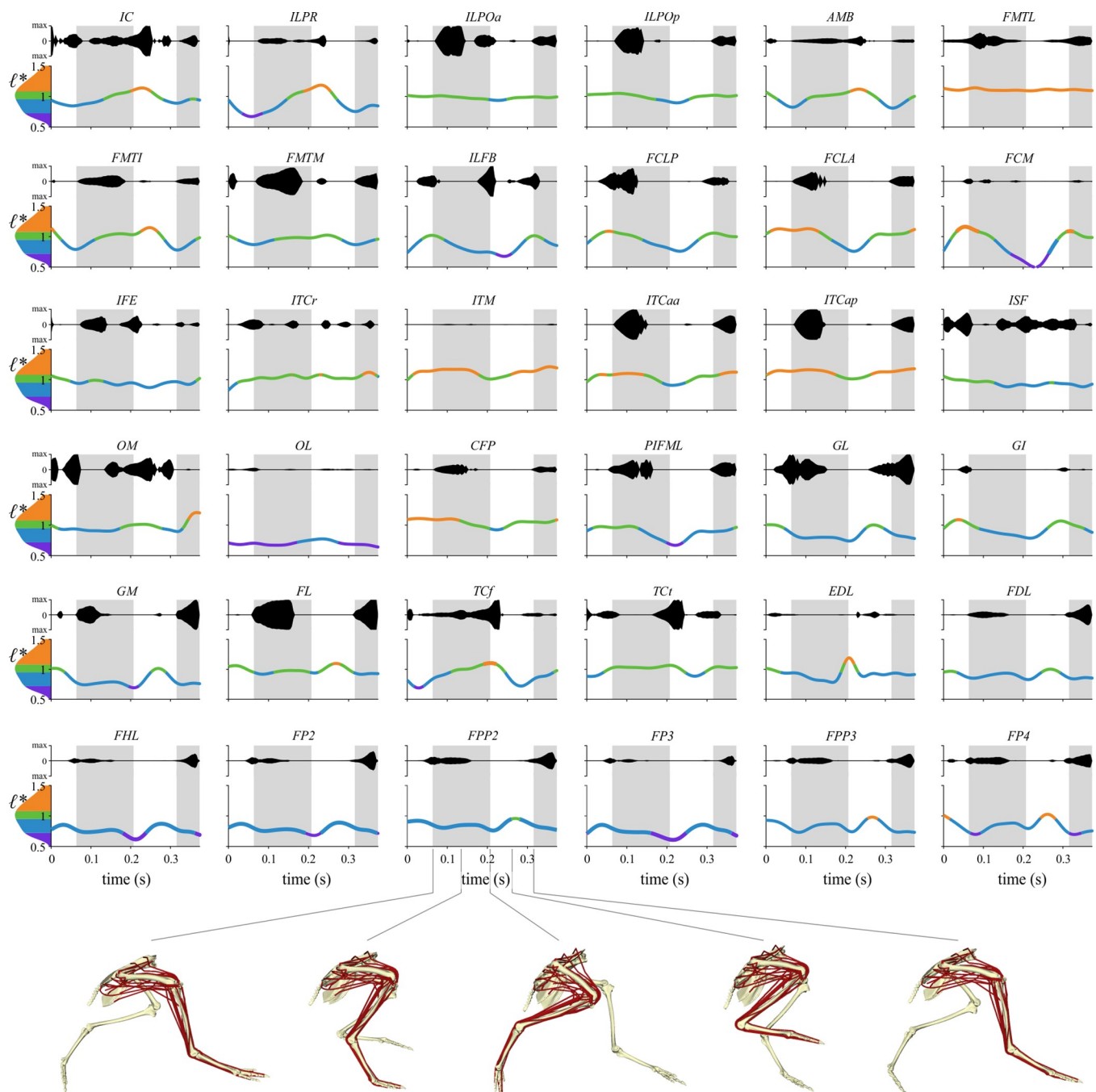

**Fig 6. Time histories of muscle excitation and fibre length change for each MTU in the running simulation.** Formatting is the same as for Fig 5.

which in some cases was not entirely consistent with published experimental data; one example (AMB) is illustrated in Fig 7. The results for a few specific muscles are worth noting in more detail. Firstly, the GL is a large, biarticular ankle extensor that has been frequently studied in birds; across all species studied to date, it is always recruited in late swing and into most of the stance phase [88]. The simulations for both walking and running recovered a consistent excitation pattern for this muscle that matches experimental observations well (Fig 7). Secondly, the

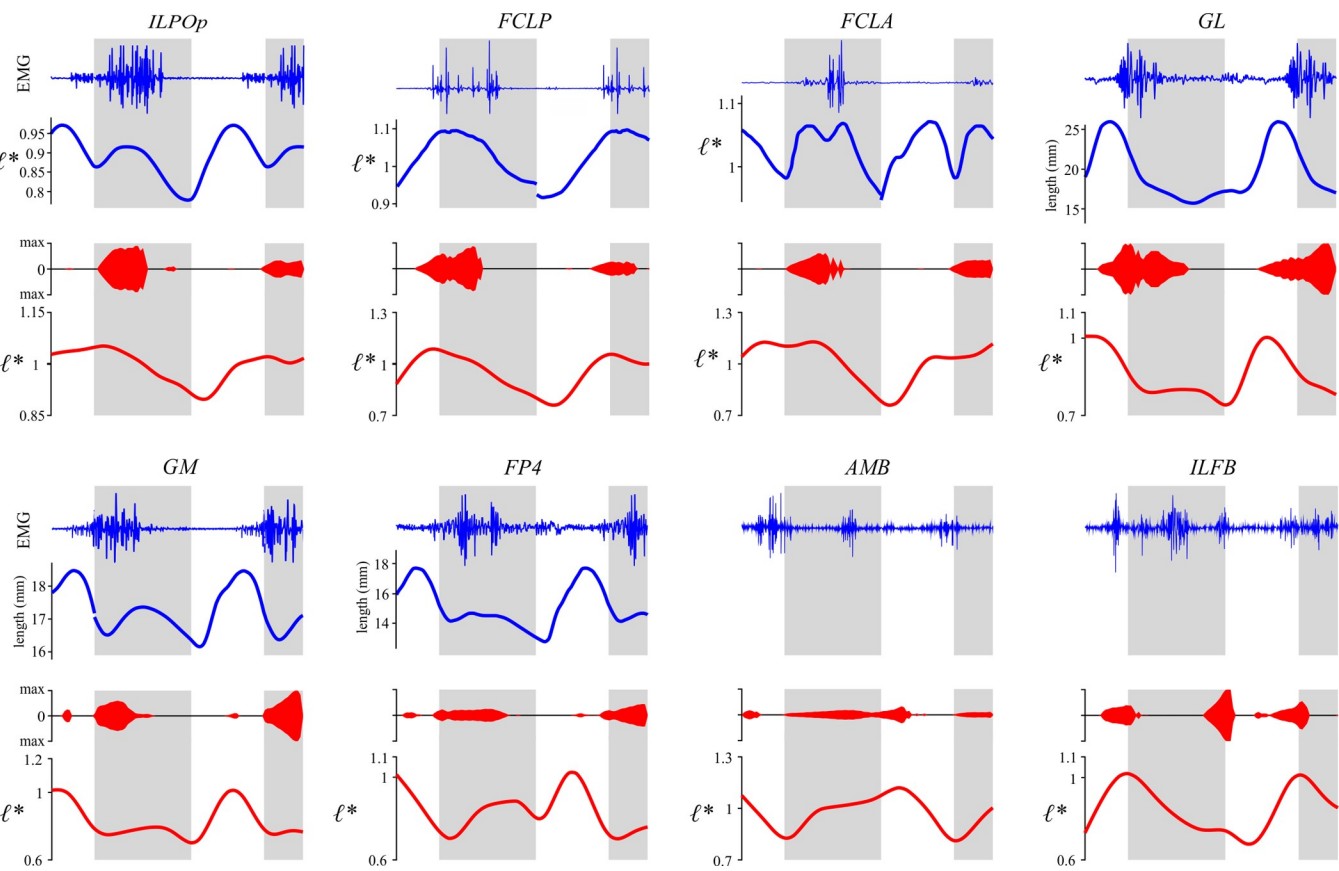

**Fig 7. Comparison of exemplar outputs for excitation and fibre length change in the running simulation against previous experimental data (electromyography and sonomicrometry), for validation of the modelling and simulation approach.** For each muscle, experimental data are shown in blue, and simulation outputs are shown in red; in some studies sonomicrometry data were reported in absolute terms, whereas in others it was reported in relative terms. See Table 2 for muscle abbreviations. Fibre length data do not exist for the AMB or ILFB. Experimental results were digitized from the original studies and scaled according to swing and stance phase durations. Sources of experimental data are as follows: ILPOp, guineafowl running at 2.5 m/s [9]; FCLP and FCLA, guineafowl running at 1.5 m/s [38]; GL, guineafowl running at 1.3 m/s [36]; GM, guineafowl running at 2 m/s [141]; FP4, guineafowl running at 1.3 m/s [36]; AMB and ILFB, guineafowl running at 1 m/s [27].

FCLA was recruited exclusively in stance phase, consistent with published experimental data [27,33,38]. This result is encouraging, because the MTU geometry in the model was distinctly different from the actual course of the muscle; in reality, the muscle originates from the belly of another muscle (FCLP) rather than the pelvis, a topology that cannot be easily incorporated in standard musculoskeletal modelling approaches. Lastly, the excitation pattern for the ILFB, with recruitment occurring almost exclusively in swing phase, was in marked contrast to published experimental data for various bird species (including tinamou), which shows it to predominantly be a stance phase muscle (Fig 7, [27,33,88,96,140]), although Marsh et al. [33] recorded activity around the stance–swing transition for anterior fibres of this muscle in guineafowl. This anomaly may suggest an incomplete description of the muscle's functional anatomy and physiological role, in either the musculoskeletal model or the OCP formulation (see also below).

There are several muscles whose recruitment during avian terrestrial locomotion remains to be experimentally investigated; combined with the results of ostrich simulations by Rankin et al. [17], the present study therefore provides important insight into the behaviour of such muscles. The FCM in the present study was negligibly recruited in the present simulations (both walking and running), whereas it was heavily recruited in the ostrich simulations,

particularly for running. This muscle is fairly large and pennate in ostriches (~0.18% of body mass [111]) but is small and parallel-fibred in the tinamou (~0.05% of body mass; Table 2), potentially explaining the difference in recovered recruitment patterns. Alternatively, such differences may reflect other genuine differences between the two species, for instance due to differing moment arms or the more than 100-fold difference in body size. In contrast to this discrepancy, the EDL and most digital flexors had similar recovered excitation patterns between the two studies; the EDL was active mostly in the swing phase, whilst the digital flexors mostly or exclusively in the stance phase, and both of these results are strongly consistent with expectations based on gross anatomy.

**Fibre operating ranges.** The time histories of fibre length changes recovered for the walking trial are presented in Fig 5, and those for the running trial in Fig 6. The patterns of change, in terms of timing of increases, decreases or stasis of fibre length with respect to stance and swing phases remained largely consistent for almost all muscles across walking and running. However, the net fibre length excursion across the stride for given muscle tended to be greater in running. Exemplar comparisons of simulated fibre length changes with previously published sonomicrometry data for walking and running birds are presented in Fig 7. However, unlike electromyography, these data have previously been collected for only a few muscles in the avian hindlimb, limiting the extent to which simulation results can be validated here.

All muscle fibres operated between $0.5 \leq \ell^* \leq 1.21$ in walking and running, with some muscles using a greater portion of this range than others. The vast majority of muscles' fibres spent most of the stride cycle on the ascending limb or the plateau of the active force–length curve. There were a few muscles whose fibres tended to operate on the descending limb for most or all of the stride cycle, which may be a consequence of the optimization combined with the initial parameters derived from dissection. Alternatively, this result may indicate that these muscles' fibres are more tuned (adapted) for other behaviours not studied here. By comparing the time histories of excitation and fibre length change, it is evident that some muscles produced force whilst fibres were undergoing contraction (concentric force production; e.g., FCLP, PIFML), others were actively producing force whilst fibres were lengthening (eccentric force production; e.g., TCf), others remained nearly isometric during force production (e.g., ITCa), and others exhibited a combination of these. This diversity of behaviours is consistent with a diversity of functional roles inferred for ostrich hindlimb muscles by Rankin et al. [17], although a detailed analysis of force–length behaviour (e.g., work loops) in the tinamou is beyond the scope of the present study.

In relation to previously recorded fascicle strain data, the time histories of fibre length changes for previously studied muscles is generally concordant with published data. In the ILPO, fibre length change across the stride was more pronounced during running than walking, and exhibited a near-monotonic decrease during the stance phase (Fig 7); this is largely consistent with published data for guineafowl, except for a small increase in length at the beginning of stance [9,41,58]. Moreover, $\ell^*$ remained on the ascending limb or plateau of the active force–length curve, also consistent with empirical observations [9,41,58]. In the FCLP, $\ell^*$ oscillated around the peak of the active force–length curve; it decreased monotonically during stance phase and continued to decrease into early swing, before increasing for the remainder of swing phase (Fig 7), a pattern experimentally observed in guineafowl [38]. Similarly, fibre length of the FCLA decreased almost monotonically throughout much of stance and increased over the swing phase (Fig 7), also consistent with experimental observations. In running, $\ell^*$ of the FCLP oscillated evenly about the peak of the active force–length curve, comparable to the results of Ellerby and Marsh [38], but in walking $\ell^*$ tended to occur more on the ascending limb. As with the excitation pattern recovered for the FCLA, this concordance between simulation and empirical observations is encouraging given the simplified representation of its MTU in the model.

The avian gastrocnemius has been extensively studied experimentally, particularly the lateral (GL) and medial (GM) heads. In the GL, fibres shortened over the whole stance phase and lengthened in the first half of swing, before shortening again in late swing (Fig 7), a pattern largely consistent with previously published data, although modest variation across species and studies is known [36,55,78,141–145]. In the simulations $\ell^*$ operated almost exclusively on the ascending limb for both walking and running, whereas published data for turkey GL indicates an operating range that includes part of the descending limb [78,143]; however, it is not certain if those studies' measures of resting fascicle length is truly representative of optimal fascicle length. Published data for guineafowl [141] and mallard ducks [142] suggest that mean $\ell^*$ also lies on the ascending limb, consistent with simulation results, but an uncertain measure of resting fascicle length was used by those studies, too. In the GM, fibre lengths underwent a small decrease across the stance phase, except for a slight increase around midstance (inverted U shape, more prominent in running), and they increased and then decreased over the course of the swing phase (Fig 7). This pattern, including the inverted U shape around midstance, is consistent with published data, especially for more proximal fibres in the muscle [41,141,145]. In contrast to the GL and GM, the gastrocnemius pars intermedia (GI) displayed a pattern of fibre length change that is not entirely consistent with limited published data for guineafowl [38], but this may in part be due to anatomical topologies (osseous and soft tissue) involving the GI and surrounding muscles that differ between guineafowl and the tinamou as modelled here [38].

More distally, only two additional avian hindlimb muscles have been experimentally investigated. In the FL, $\ell^*$ remained fairly constant over the stride in walking, operating on the ascending limb of the active force–length curve; however, in running, $\ell^*$ showed an increase-then-decrease pattern during the swing phase, oscillating from ascending to descending limbs, a pattern somewhat consistent with published data [78,143]. In the FP4 (a digital flexor), fibres increased in length over the stance phase followed by a lengthening-then-shortening pattern in swing; this pattern was more exaggerated in running, and is partially consistent with data reported for guineafowl [36,144].

## Static simulations

In total, there were 3,827 viable poses for simulations 1 and 2 (76.5% of all tested), and 4,461 viable poses for simulations 3 and 4 (89.2% of all tested). An attempt was also made to perform the static simulations using the original (dissection-derived) values for $\ell_o$ and $L_S$, but only 221 viable poses for simulations 1 and 2, and 4,586 viable poses for simulations 3 and 4, were recovered; this supports the use of MTU tuning in the solving of the inverse simulations.

The distributions and ranges of normalized fibre length for each MTU were very similar across the four analyses, and so only the results for simulation 1 are presented in Fig 8. Viable operating ranges varied considerably among the MTUs, with some (e.g., FMTM, OL, TCt) exhibiting a much narrower range of $\ell^*$ than others (e.g., ILPR, FCM, FP2). Importantly, few muscles had a viable operating range approximately equal to optimal fibre length, and moreover few muscles had a fibre operating range approximating $0.5 \leq \ell^* \leq 1.5$. Six muscles' fibres had a lower bound on their operating range that was significantly less than 0.5, which would be unlikely to be used during *in vivo* behaviours as it corresponds to near-zero MTU force (but is nevertheless included here for the sake of completeness). Only one muscle (EDL) had fibres that exhibited an operating range beyond 1.5 in any of the simulations. Superimposing the *in vivo* range of fibre lengths recovered for the walking and running trials shows that most muscle fibres used only a fraction of their total viable operating range during locomotion (Fig 8; median of 25.5% for walking, 44.8% for running).

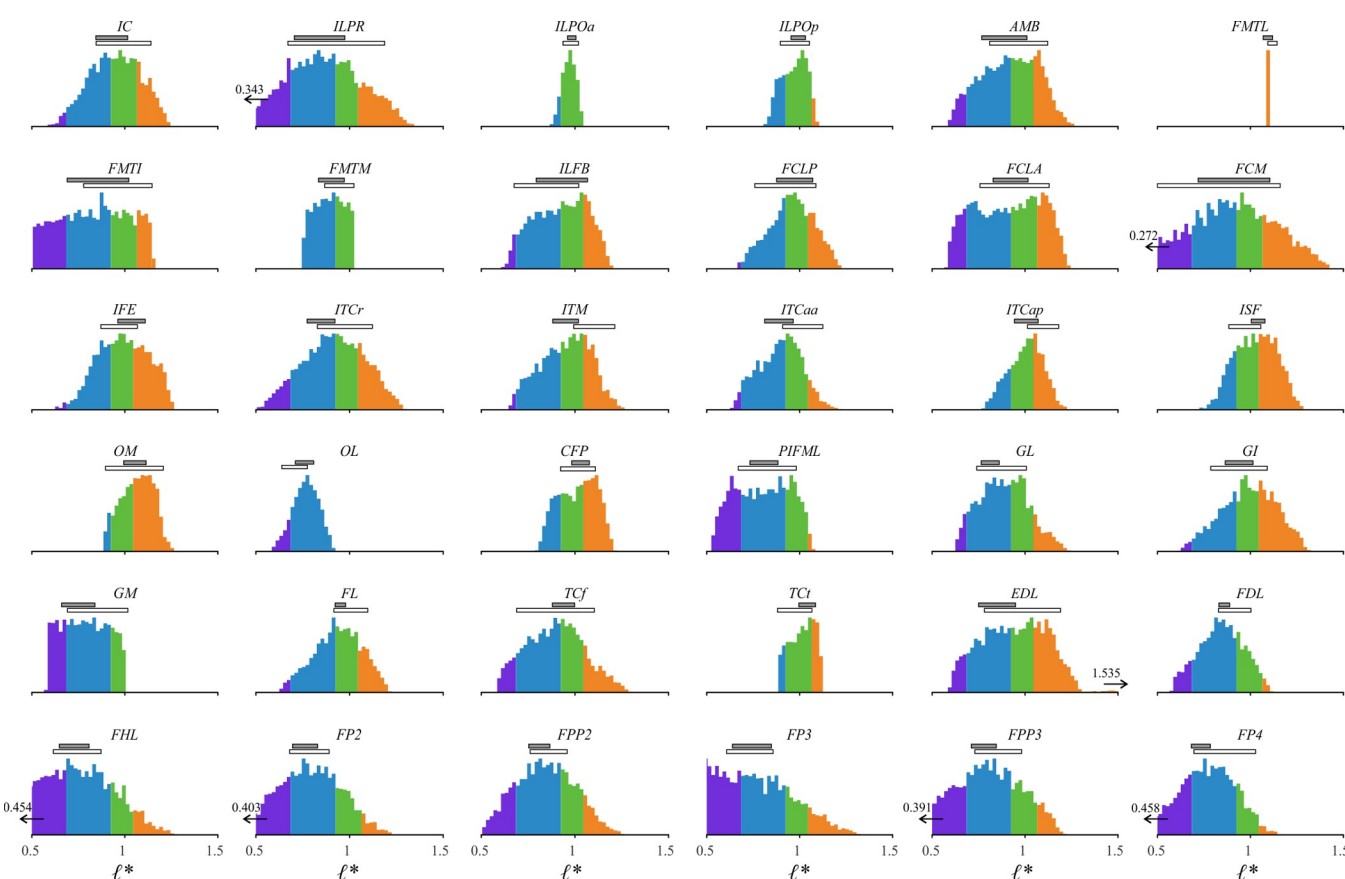

**Fig 8. Distribution of normalized fibre lengths for each MTU in static simulation 1 (3,827 viable poses).** Each distribution is visualized as a histogram (intervals of 0.02× $\ell_o$), the vertical axis of which is scaled by the magnitude (number of counts) of the most frequent bin. Colours signify the various parts of the active force–length curve as per Figs 5 and 6. Also shown are the ranges of normalized fibre lengths used *in vivo* during walking (grey bars) and running (white bars). Instances where the minimum or maximum normalized fibre length achieved reached below 0.5 or above 1.5 (respectively) are indicated. The very small range recovered for the FMTL is due to the way its path was represented in the musculoskeletal model, crossing over the lateral femoral condyle close to the flexion–extension axis of the knee, such that it had little opportunity to undergo change in MTU length in the first instance. Similarly, the small range recovered for the ILPOa likewise reflects its path in the model passing close to the flexion–extension axes of the hip and knee. See Table 2 for muscle abbreviations.

Variation in the observed ranges of normalized fibre length was mirrored by variation in the ranges of $l_{MT}$, although there was by no means a strict correlation between the two (Fig 9A). Notably, very few MTUs experienced a total length change approximating $\ell_o$, in any of the simulations (the same result was also obtained in static simulations using the un-tuned values for $\ell_o$ and $L_S$). A greater range in $l_{MT}$ across the tested limb poses generally corresponded with a greater range in $\ell^*$, with the latter being less than the former, sometimes markedly so. The discrepancy between $l_{MT}$ range and $\ell^*$ range was found to be explained by both muscle pennation angle and the relative length of tendon in the MTU (computed as the ratio of $L_S$ to $\ell_o$), as determined by ordinary least squares regression in PAST 3.09 [146]. A greater decoupling between fibre and MTU length change results from greater pennation (Fig 9B; $r^2 = 0.639$, $P < 0.001$), or a relatively longer tendon (i.e., greater potential for tendon stretch to contribute toward $l_{MT}$ change; Fig 9C; $r^2 = 0.655$, $P < 0.001$).

Comparing $\ell^*$ to joint angles revealed a variety of relationships, representative examples of which are illustrated in Fig 10. Hyperplane fitting (Table 4) indicates that for some muscles the relationship between $\ell^*$ and joint angle(s) was more linear, or proportional, than for others:

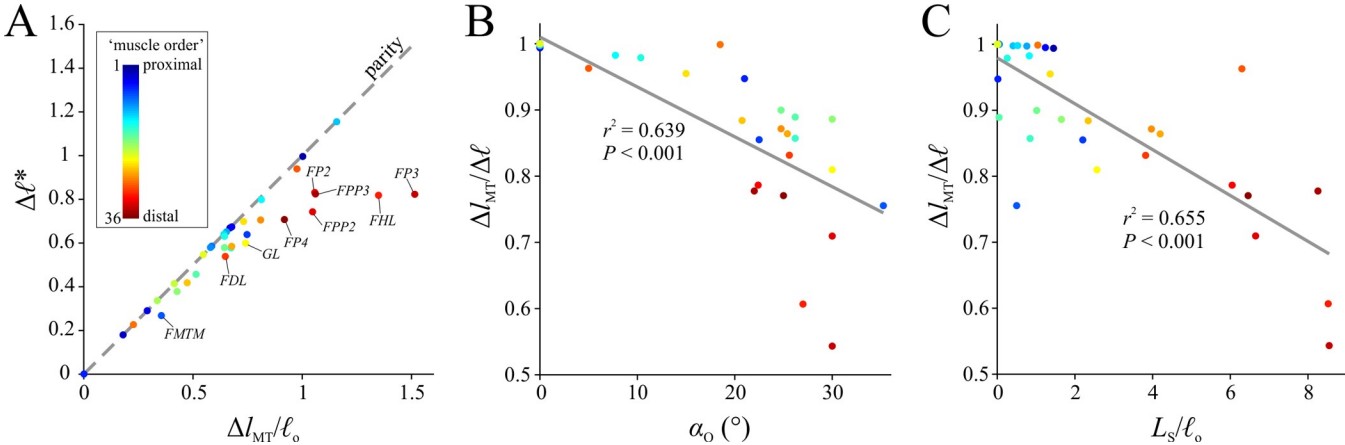

**Fig 9. Ranges of normalized fibre lengths and MTU lengths across all viable limb poses.** The results illustrated here are for static simulation 1, but a very similar pattern was achieved for the other three simulations as well. (A) Range of normalized fibre length compared to range of MTU length (normalized by optimal fibre length) for each MTU; all points plot on or below the line of parity, with the most extreme outliers indicated (see Table 2 for abbreviations). (B) Ratio of MTU length range to fibre length range compared to pennation angle. As fibre length range can at most be equal to MTU length range, this ratio is one or less; the lower it is (i.e., the further a point plots further from the line or parity in A), the greater the discrepancy between length changes at the level of the MTU and fibre. A higher pennation angle leads to a greater decoupling between changes in fibre and MTU lengths. (C) Ratio of MTU length range to fibre length range compared to relative tendon length, expressed as the ratio of slack to fibre length. A relatively longer tendon leads to a greater decoupling between changes in fibre and MTU lengths. In all panels, points are colour coded by the order of muscles as listed in Table 2, which are arranged in a proximal-to-distal fashion.

59.7% of muscle–simulation (simulations 1+2, or 3+4) combinations had $r^2 > 0.95$, and 80.6% had $r^2 > 0.90$; 11.1% of combinations had RMSE $> 5\%$ of $\ell_o$; and 54.2% of combinations had $\varepsilon_{max} > 10\%$ of $\ell_o$, and 15.3% had $\varepsilon_{max} > 20\%$ of $\ell_o$.

## Discussion

This study principally sought to investigate how muscle fibre length varies during the execution of locomotor behaviour in a previously little-studied species of ground bird. Fibre mechanics play an integral role in the ability of muscles to confer movement, and ground-dwelling bird species are frequently used as a model system for understanding musculoskeletal function in bipedal locomotion [28,29,44]. Yet experimental constraints render the investigation of *in vivo* fibre mechanics difficult, such that most avian hindlimb muscles have hitherto remained underexplored, if at all. Through *in silico* dynamic simulations that integrated high-quality experimental data and musculoskeletal modelling, the present study marks the first attempt to investigate *in vivo* fibre length changes in all important muscles of the avian hindlimb at once. A key overarching result was that the fibres of all MTUs operated between $0.5 \leq \ell^* \leq 1.21$ in walking and running, with most spending the majority of their time on the ascending limb or plateau of the active force–length curve, thus answering one of the study's main questions. This finding is consistent with a variety of modelling and experimental studies in humans [e.g., 8,15,51,76,81], as well as limited published experimental data for birds [9,38,41,56,58,78,141–143].

The high-fidelity tinamou musculoskeletal model developed here adds to the diversity of avian species for which such models exist [46,47,49,50,111], helping to facilitate more detailed future studies of interspecific variation in (and evolution of) musculoskeletal function and performance. The tinamou model was used in the present study to explore just one aspect of musculoskeletal function in straight, steady-state locomotion, but in reality is easily amenable to investigating other aspects of musculoskeletal function and for diverse other behaviours, such as acceleration [147], turning [148], or sit-to-stand manoeuvres [18]. Indeed, in concert with predictive

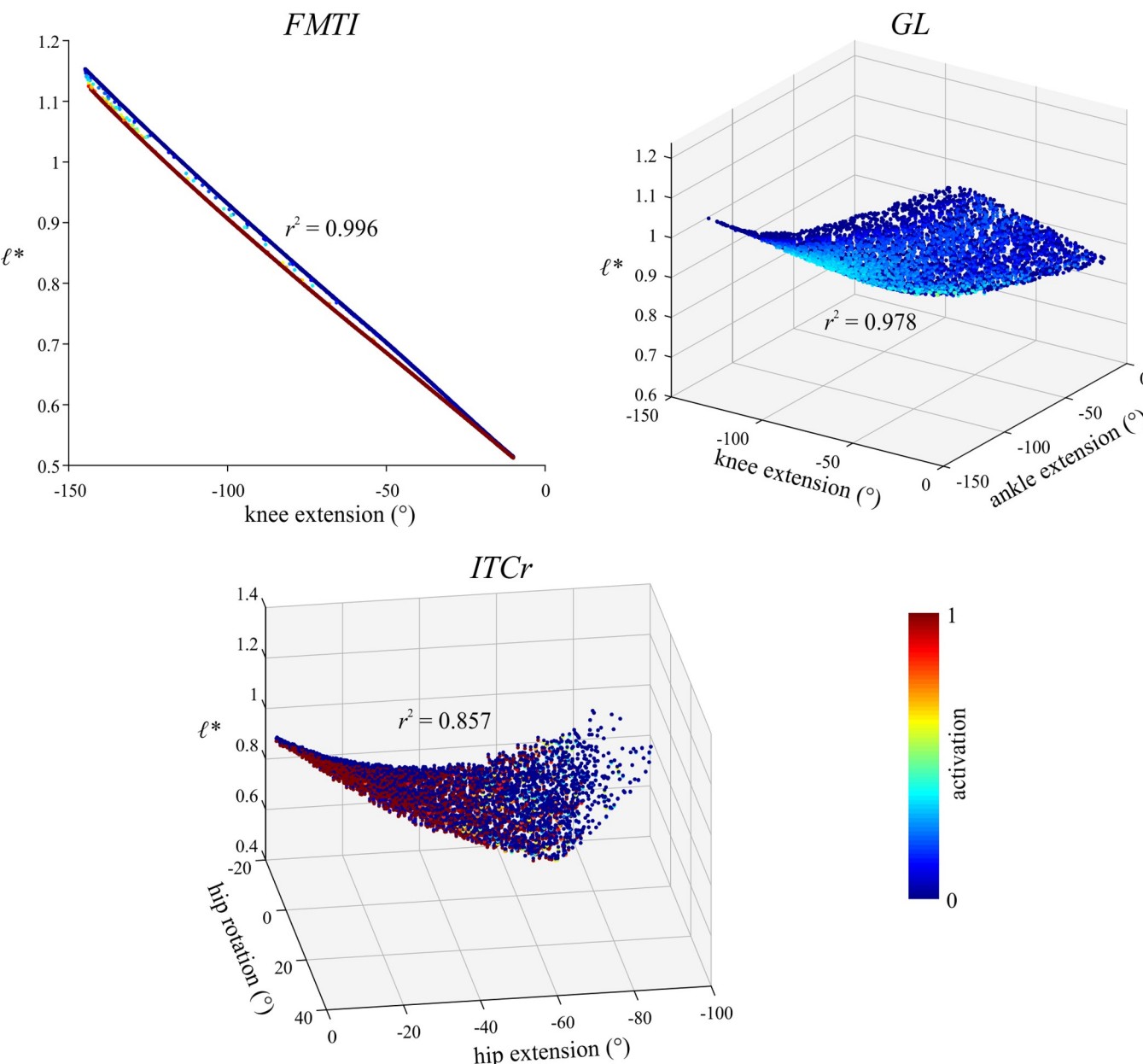

**Fig 10. Exemplar plots illustrating variation in normalized fibre length with respect to joint angle(s) in the static simulations, shown here for the combined results of simulations 1 and 2.** The FMTI actuates one DOF, the GL actuates two DOFs and the ITCr actuates three DOFs (only two of which are plotted here; hip abduction omitted for clarity). The coefficient of determination for a hyperplane fit is also given in each case. See Table 2 for muscle abbreviations.

simulation techniques, the model is currently being used to explore muscle function during the execution of jumping and landing behaviour (P.J. Bishop et al., in review). It is hoped that in turn a clearer and more rounded understanding of avian musculoskeletal function will be achieved.

## MTU tuning in inverse simulations

In a similar fashion to previous work [22,135], this study has also extended the sophistication of the Muscle Redundancy Solver package [7] used for solving inverse simulations. In addition

**Table 4. Results for hyperplane fitting to investigate the linearity (proportionality) of the relationship between normalized fibre length and joint angle(s) for each MTU.** Fitting was applied to the results from simulations 1 and 2 together, and to the results from simulations 3 and 4 together. RMSE and $\varepsilon_{max}$ are reported with respect to optimal fibre length. In addition to the results for each muscle individually, the grand mean and standard deviation across all muscles is reported.

| | simulations 1 and 2 | | | simulations 3 and 4 | | |
|---|---|---|---|---|---|---|
| | $r^2$ | RMSE | $\varepsilon_{max}$ | $r^2$ | RMSE | $\varepsilon_{max}$ |
| **overall** | **0.925 ± 0.14** | **0.025 ± 0.017** | **0.117 ± 0.099** | **0.916 ± 0.147** | **0.028 ± 0.018** | **0.123 ± 0.117** |
| IC | 0.9729 | 0.0221 | 0.1684 | 0.9603 | 0.0276 | 0.1444 |
| ILPR | 0.9889 | 0.0211 | 0.1481 | 0.9859 | 0.0251 | 0.1362 |
| ILPOa | 0.9471 | 0.0081 | 0.0344 | 0.9374 | 0.0088 | 0.0311 |
| ILPOp | 0.9765 | 0.0101 | 0.0488 | 0.9699 | 0.0117 | 0.0442 |
| AMB | 0.9383 | 0.0372 | 0.1766 | 0.9229 | 0.0418 | 0.1484 |
| FMTL | 0.2996 | 0.0002 | 0.0005 | 0.2485 | 0.0002 | 0.0005 |
| FMTI | 0.9960 | 0.0112 | 0.0215 | 0.9962 | 0.0109 | 0.0242 |
| FMTM | 0.9988 | 0.0026 | 0.0059 | 0.9988 | 0.0026 | 0.0076 |
| ILFB | 0.9907 | 0.0133 | 0.0428 | 0.9719 | 0.0238 | 0.5898 |
| FCLP | 0.9895 | 0.0120 | 0.0556 | 0.9893 | 0.0129 | 0.0569 |
| FCLA | 0.9895 | 0.0178 | 0.0928 | 0.9880 | 0.0199 | 0.0917 |
| FCM | 0.9944 | 0.0180 | 0.1028 | 0.9946 | 0.0189 | 0.1012 |
| IFE | 0.8118 | 0.0578 | 0.3085 | 0.7695 | 0.0634 | 0.3108 |
| ITCr | 0.8574 | 0.0613 | 0.2853 | 0.8104 | 0.0694 | 0.2697 |
| ITM | 0.9262 | 0.0356 | 0.1408 | 0.9187 | 0.0384 | 0.1386 |
| ITCaa | 0.9477 | 0.0263 | 0.1000 | 0.9465 | 0.0278 | 0.1097 |
| ITCap | 0.9778 | 0.0134 | 0.0503 | 0.9778 | 0.0134 | 0.0503 |
| ISF | 0.9202 | 0.0311 | 0.1221 | 0.9202 | 0.0311 | 0.1221 |
| OM | 0.4882 | 0.0603 | 0.2574 | 0.4882 | 0.0603 | 0.2574 |
| OL | 0.8526 | 0.0265 | 0.1011 | 0.8526 | 0.0265 | 0.1011 |
| CFP | 0.9834 | 0.0126 | 0.0621 | 0.9834 | 0.0126 | 0.0621 |
| PIFML | 0.9906 | 0.0138 | 0.0794 | 0.9906 | 0.0138 | 0.0794 |
| GL | 0.9777 | 0.0193 | 0.0637 | 0.9748 | 0.0215 | 0.0751 |
| GI | 0.9871 | 0.0164 | 0.0447 | 0.9868 | 0.0167 | 0.0451 |
| GM | 0.9992 | 0.0033 | 0.0113 | 0.9911 | 0.0116 | 0.0200 |
| FL | 0.9955 | 0.0084 | 0.0345 | 0.9704 | 0.0229 | 0.0479 |
| TCf | 0.8962 | 0.0481 | 0.1471 | 0.8917 | 0.049 | 0.1497 |
| TCt | 0.9901 | 0.0063 | 0.0123 | 0.9878 | 0.0073 | 0.0215 |
| EDL | 0.8972 | 0.0543 | 0.3993 | 0.8863 | 0.0571 | 0.3923 |
| FDL | 0.9926 | 0.0103 | 0.0588 | 0.9735 | 0.0204 | 0.0646 |
| FHL | 0.9565 | 0.0390 | 0.2430 | 0.9497 | 0.0440 | 0.1430 |
| FP2 | 0.9796 | 0.0242 | 0.1076 | 0.9587 | 0.0354 | 0.1124 |
| FPP2 | 0.9644 | 0.0295 | 0.1195 | 0.9477 | 0.0375 | 0.1090 |
| FP3 | 0.9338 | 0.0489 | 0.3283 | 0.9537 | 0.0442 | 0.1538 |
| FPP3 | 0.9683 | 0.0337 | 0.1118 | 0.9514 | 0.0428 | 0.1080 |
| FP4 | 0.9134 | 0.0420 | 0.1190 | 0.9145 | 0.0432 | 0.1239 |

to being able to solve for multiple trials simultaneously, the capability now exists to tune muscle architectural parameters 'on the fly' whilst excitations and MTU forces are being solved for. Such tuning can accommodate measurement or modelling errors that may otherwise hinder a musculoskeletal model's ability to execute a recorded behaviour without excessive reliance on reserve actuators. It therefore affords the possibility of correcting for errors that can creep in at previous stages of the experimental or modelling workflow, such that spurious conclusions

may potentially be avoided. The ability to tune also presents the future opportunity of solving and tuning for specific behaviours one-at-a-time, thereby shedding insight on how muscle–tendon anatomy may need to be adapted for the execution of a specific behaviour, how this may affect the execution of a different behaviour, and how trade-offs may exist for some muscles across disparate behaviours or performance requirements [149,150]. In turn, this tuning approach can provide the foundation to reconciling interspecific differences in muscle architecture (or more generally, musculoskeletal function) in terms of differences in ecology and selective regime [e.g., 151,152,153].

Of greater immediate relevance, however, is the tuning that took place in the present study. Only optimal fibre length ($\ell_o$) and tendon slack length ($L_S$) were tuned here; whereas prior simulation studies have sometimes used increased values of $F_{max}$ (or equivalently, maximum stress [e.g., 80,117,154,155]), this parameter was left unaltered in the present study. It is therefore encouraging that even in the more strenuous running trial the model was able to reproduce the recorded behaviour with little reliance on reserve actuators. Almost all MTUs needed both $\ell_o$ and $L_S$ to be increased from their original values, but the proximate reason for this apparent consistency remains unclear. It may reflect systematic error in measurement of architectural parameters during dissection, the way in which a Hill model effectively represents an entire muscle as a single fibre [156], error in the assumption that measured fascicle lengths correspond to optimal fibre lengths, erroneous values for assigned tendon stiffness, muscle-specific variation in physiology (e.g., preponderance of slow- *versus* fast- twitch fibres, affecting fibre contraction velocity), or a combination of these and other factors. Nevertheless, many MTUs in the tinamou model required a relatively minor amount (~10%) of tuning of the original values for $\ell_o$ (measured) and $L_S$ (estimated). By the same token, however, this small amount may have significant consequences for the capabilities of the musculoskeletal model. For example, the all-muscle static simulations (1 and 2) showed that MTUs with the original parameters could not coordinate a viable solution for the vast majority of the 5,000 limb poses tested, highlighting the nonlinear relationships between muscle architecture and force-producing capability.

### Inverse simulation results

In addition to reconstructing muscle excitation and fibre length changes for all key muscles of the tinamou hindlimb, the results obtained here provide new perspective on *in vivo* fibre length changes in muscles previously studied only in neognath birds, as noted above. Furthermore, recovered patterns of muscle recruitment in walking and running were largely consistent with previously published electromyographic data; this includes, encouragingly, the FCLA, whose path in the musculoskeletal model is considerably different in topology compared to reality. These results further bolster the inference of broad conservatism in muscle recruitment strategies across extant birds [88]. The present study is the first to investigate the tiny obturatorius lateralis (OL), which was minimally excited in both walking and running. However, its low recruitment here may be more a consequence of the very small force-generating capacity of this muscle, rather than a biologically meaningful result; with a low $F_{max}$ (Table 2), the production of any meaningful amount of force would require a sizeable activation, which would be penalized in the current formulation of the objective function (Eq 4). Future work could explore modification of the activation term in the objective function that accounts for disparity in muscle size across the limb [157].

The greatest discordance between the results of inverse simulation and previous experimental observation is the behaviour of the iliofibularis (ILFB; Fig 7), a large, biarticular hip extensor and knee flexor analogous to the mammalian biceps femoris. In the simulations it was recruited almost exclusively in the swing phase, whereas electromyographic recordings of chickens,

guineafowl and emus [27,33,88,96,140], and inverse simulations of ostrich gait [17], consistently show it to be predominantly active during stance. During the swing phase, this muscle's fibres undergo net lengthening (Fig 7; no experimental sonomicrometry data exist), indicating that in the current simulations the muscle's fibres were eccentrically contracting, acting as a brake. The ILFB has the largest knee flexion moment arm of any MTU in the model, but programmatic reduction of its flexion moment arm during the simulations did not result in a change to stance-dominant recruitment activity until moment arms were reduced by a factor of 10, which is considered highly implausible. Clearly this result warrants future investigation, one avenue of which could be modification of the force–velocity curve used in this study [7], which may currently be too 'generous' on its lengthening side. Alternatively, it is possible that the current objective function used does not capture all salient imperatives dictating muscle recruitment strategy. For instance, recruitment of the ILFB, and perhaps other muscles, may be influenced by afferent inputs [e.g., 158], yet sensory feedback was not a component of these simulations.

One final result from the inverse simulation worth noting is the recruitment of reserve actuators. Although the recommended threshold of 5% of external joint moment suggested by Hicks et al. [118] for human studies was not always met throughout the entirety of the simulations, by and large the reserves contributed relatively little effort toward driving the model's motion. However, it was necessary to permit the reserves actuating the abduction–adduction and long-axis rotation DOFs of the knee and ankle to be much more strongly recruited. As noted in the Results, lower recruitment of these reserves produced implausible results insofar as many distal muscles were concerned. Moreover, re-running the inverse simulations with these four DOFs excluded (but kinematics remaining otherwise unaltered) produced excitation and fibre length results that were extremely similar to the results presented above. Collectively, this is interpreted to mean that these four DOFs in life receive a much greater contribution to joint moment balance from passive structures, such as ligaments, the joint capsule and bony articular surface geometry. Indeed, cadaveric manipulations of de-fleshed cadavers shows that the avian knee and ankle are both largely restricted to flexion-extension, with movement about other axes more constrained by soft tissues and articular geometry ([93,159], pers. obs. of many avian species, including tinamous). The strong reliance on reserves for controlling these DOFs in the present simulation is therefore considered justifiable. In the ostrich simulations of Rankin et al. [17], it was found that the hip abduction–adduction reserve actuator was quite strongly recruited (activation exceeding 50%), which was further suggested to reflect the action of largely passive structures at the hip. The level of recruitment for the same reserve actuator in the present tinamou simulation was lower throughout walking or running, but nonetheless was typically the highest among all the DOFs (except those noted above), again suggesting a possibly important stabilizing role of passive structures in the avian hip, such as the bony antitrochanter and hip joint ligaments [see also 139]. The relevance of passive structures to moment balance in avian joints during movement could be more precisely investigated in the future by explicitly incorporating a representation of such structures into the optimal control problem. That is, rather than using abstract torque actuators to provide non-muscular forces (as done here), a mechanistic model of passive force generation could be implemented, one which can at least theoretically act in an energy conservative manner. Solving the optimal control problem would likely also require a finer problem discretization than that used here, due to the way in which dynamics are approximated in the numerical approach employed.

## Static simulation results

A secondary goal of this study was to investigate how muscle fibre lengths vary throughout the entire range of potential hindlimb motion in the tinamou, providing bearing on a key

assumption previously used to estimate muscle parameters in extinct species [66,73,74]. This was achieved through performing a series of basic static simulations with the tuned musculoskeletal model over the full (indeed, more than physiologically possible) range of limb postures. The static simulations showed that by and large the tuned MTUs were still able to function over the range of limb poses tested, suggesting that the tuning process did not produce a strong postural bias in MTU functionality. However, the simulations revealed that few muscle fibres had an operating range approximately equal to $\ell_o$, and moreover fibre operating ranges rarely approximated $0.5 \leq \ell^* \leq 1.5$. Furthermore, the total change in $l_{MT}$ across the postures tested rarely approximated $\ell_o$, with most MTUs experiencing a net length change less than this value. This implies that the practice of using $l_{MT}$ change to estimate $\ell_o$ in extinct species will tend to produce underestimates of $\ell_o$ for many muscles, and in turn overestimates of these muscles' force-producing capacity (Eq 1) or underestimates of their viable fibre operating range. It remains to be demonstrated how much this misestimation could affect performance estimates for extinct species, such as maximal running speed, or whether the hindlimb muscles of giant species (e.g., tyrannosaurid dinosaurs) could sustain any form of running gait at all [59,66].

Hyperplane fitting of $\ell^*$ against joint angle(s) demonstrated that it cannot by default be assumed that $\ell^*$ varies in strict proportion to joint angle(s). Certainly for some muscles a strong linear relationship existed between the two, but calculated $r^2$ and RMSE indicate that, in the tinamou at least, this is not always the case. This is not at all surprising, as moment arms for many muscles can vary considerably across joint motion (immediately invalidating an assumption of strict proportionality); moreover, muscle–tendon compliance and biarticularity can further complicate the issue [50,74,82]. Values of $r^2$, RMSE and $\varepsilon_{max}$ are presented here for each muscle (Table 4), but it is left to the reader to decide what level of error (deviation from the assumption of proportionality) is acceptable for their particular research purposes. Given the dramatic difference in pose viability between using original and tuned values for $\ell_o$ (and $L_S$) noted above, the fact that many MTUs had $\ell_o$ tuned by <10%, and the fact that $\varepsilon_{max}$ for many MTUs was sizeable, the results presented here emphasize the importance of accurate estimation of $\ell_o$, urging caution regarding simplistic assumptions.

## Conclusions and future directions

Using high-quality experimental kinematic and kinetic data with a sophisticated musculoskeletal modelling and simulation approach, fibre length changes in every key muscle of the avian hindlimb were investigated for the first time. It is important to recognize, however, that the limited amount of suitable experimental data did not permit an assessment of stride-to-stride or inter-individual variation of fibre behaviour in the present study, allowing only a coarse assessment of how fibre behaviour changed with speed. Moreover, as with many previous studies, the current investigation was limited to walking and slow running, which may have influenced the tuning process. A key goal for future modelling studies, then, is to explore how hindlimb muscle fibre mechanics (and more broadly, musculoskeletal function) varies with other behaviours, such as acceleration, deceleration, turning, sit-to-stand manoeuvres and jumping. Only through investigating a wide variety of ecologically relevant behaviours can the constraints on hindlimb musculoskeletal anatomy, function and performance in birds be truly understood.

Paralleling the findings of previous studies, it was found here that muscle fibres used only a fraction of their active force–length curve, tending to spend most of their time on the ascending limb or near-isometric plateau. This is consonant with the conservative structure of vertebrate skeletal muscle [1], suggesting functional conservatism in an evolutionary context. Such

conservatism encourages the development of overarching principles relating muscle–tendon architecture to skeletal anatomy and joint mobility. The results of the present study can help refine such principles, which in tandem with anatomical and functional studies of a broader range of extant species, can be applied (with caution) to estimating muscle architecture and function in the fossil record.

## Supporting information

**S1 Fig. Identification of individual muscle–tendon units in the tinamou musculoskeletal model (see Table 2 for abbreviations).** (A) Iliotibiales. (B) Deep knee extensors. (C) 'Hamstrings'. (D) Deep external rotators. (E) Deep internal rotators. (F) Caudal hip extensors. (G) Gastrocnemii. (H). Ankle flexors. (I) Ankle extensors and metatarsophalangeal plantarflexors; the digital flexors are the FDL, FHL, FP2, FPP2, FP3, FPP3 and FP4.
(TIF)

**S2 Fig. Comparison of the experimentally recorded ground reaction forces to those obtained in the tracking simulation, shown for the walking trial.** The running trial is not shown since no experimental data was able to be collected here.
(TIF)

**S1 Table. Segment-specific densities measured in individual DDT07 and used in the tinamou model.**
(XLSX)

**S1 Movie. Animation of walking tinamou simulation, showing kinematics, ground reaction forces and reconstructed muscle excitation patterns for the left hindlimb, played at 0.1× real speed.** Also shown are the forceplates used to collect ground reaction data in the global coordinate system, and the location of the instantaneous whole-body centre of mass. Muscles become redder as they become more excited.
(MP4)

**S2 Movie. Animation of running tinamou simulation, showing kinematics, (simulated) ground reaction forces and reconstructed muscle excitation patterns for the left hindlimb, played at 0.1× real speed.** Also shown is the plane of the treadmill in the global coordinate system, and the location of the instantaneous whole-body centre of mass. Muscles become redder as they become more excited.
(MP4)

**S1 Model File. The final (tuned) OpenSim musculoskeletal model, along with input kinematics and kinetics files that can be imported directly into OpenSim.**
(ZIP)

**S1 Code MATLAB code used for (a) processing raw forceplate data, (b) performing the tracking simulations, (c) performing the inverse simulation, and (d) performing the static simulations.**
(ZIP)

## Acknowledgments

The following people are sincerely thanked for their contributions throughout various phases of this study: E. Sparkes and T. West for assistance in the setup of experiments and collection of data, staff of the Biological Services Unit at the Royal Veterinary College for animal care and surgical support, C. Adami, H. Ronaldson, P. Monticelli and L. Pelligand for planning and

conducting tinamou anaesthesia, J. Rankin for insight with musculoskeletal modelling, simulation and optimization, S. Gatesy, A. Manafzadeh, D. Baier and B. Knörlein for assistance with XROMM, E. Herbst for assistance with Maya software, K. Smithson, V. Watts and A. Moors for CT scanning, and colleagues in the Structure and Motion Laboratory for helpful discussion. The helpful comments of A. Manafzadeh on earlier drafts of the manuscript are also much appreciated.

## Author Contributions

**Conceptualization:** Peter J. Bishop, Krijn B. Michel, Antoine Falisse, Andrew R. Cuff, Vivian R. Allen, Friedl De Groote, John R. Hutchinson.

**Data curation:** Peter J. Bishop.

**Formal analysis:** Peter J. Bishop.

**Funding acquisition:** Friedl De Groote, John R. Hutchinson.

**Investigation:** Peter J. Bishop, Krijn B. Michel, Antoine Falisse, Andrew R. Cuff, Vivian R. Allen, Friedl De Groote, John R. Hutchinson.

**Methodology:** Peter J. Bishop, Krijn B. Michel, Antoine Falisse, Andrew R. Cuff, Vivian R. Allen, Friedl De Groote, John R. Hutchinson.

**Project administration:** John R. Hutchinson.

**Resources:** Peter J. Bishop, Krijn B. Michel, Andrew R. Cuff, John R. Hutchinson.

**Software:** Peter J. Bishop, Antoine Falisse, Friedl De Groote.

**Validation:** Peter J. Bishop, Antoine Falisse.

**Visualization:** Peter J. Bishop.

**Writing – original draft:** Peter J. Bishop.

**Writing – review & editing:** Peter J. Bishop, Krijn B. Michel, Antoine Falisse, Andrew R. Cuff, Vivian R. Allen, Friedl De Groote, John R. Hutchinson.

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
