## [Decision Letter · Decision Letter 0]

30 Sep 2020

Dear Dr. Bishop,

Thank you very much for submitting your manuscript "Computational modelling of muscle fibre operating ranges in the hindlimb of a small ground bird (Eudromia elegans), with implications for modelling locomotion in extinct species" for consideration at PLOS Computational Biology.

As with all papers reviewed by the journal, your manuscript was reviewed by members of the editorial board and by several independent reviewers. In light of the reviews (below this email), we would like to invite the resubmission of a significantly-revised version that takes into account the reviewers' comments.

The reviewers have provided detailed suggestions for revisions and have asked for clarifications on a few points. We look forward to receiving a revised manuscript.

We cannot make any decision about publication until we have seen the revised manuscript and your response to the reviewers' comments. Your revised manuscript is also likely to be sent to reviewers for further evaluation.

Sincerely,

Manoj Srinivasan, Ph.D.

Guest Editor

PLOS Computational Biology

Stefano Allesina

Deputy Editor

PLOS Computational Biology

The reviewers have provided detailed suggestions for revisions and have asked for clarifications on a few points. We look forward to receiving a revised manuscript.

Reviewer's Responses to Questions

**Comments to the Authors:**

Reviewer #1: Nice paper, see attachment.

Reviewer #2: please find the review attached

**Have all data underlying the figures and results presented in the manuscript been provided?**

Reviewer #1: Yes

Reviewer #2: Yes

PLOS authors have the option to publish the peer review history of their article (what does this mean?). If published, this will include your full peer review and any attached files.

Reviewer #1: No

Reviewer #2: **Yes: **Koen Lemaire
---

## [Decision Letter · Decision Letter 1]

11 Dec 2020

Dear Dr. Bishop,

Thank you very much for submitting your manuscript "Computational modelling of muscle fibre operating ranges in the hindlimb of a small ground bird (Eudromia elegans), with implications for modelling locomotion in extinct species" for consideration at PLOS Computational Biology. As with all papers reviewed by the journal, your manuscript was reviewed by members of the editorial board and by several independent reviewers. The reviewers appreciated the attention to an important topic. Based on the reviews, we are likely to accept this manuscript for publication, providing that you modify the manuscript according to the review recommendations.

Both reviewers were mostly satisfied with the revisions, but both reviewers, each, indicate a disagreement or a misunderstanding with respect to one of their comments/suggestions. We look forward to your addressing these issues and clearing up the misunderstandings in the next revision. One of the reviewers also suggests other minor revisions.

I agree with the reviewer that it may make sense to remark on the GRM modifications or comparison, and do not see why the authors say that such comparison "is not meaningful". Also, isn't it is not technically true that feet can only produce moments against the ground in the vertical direction: it's just that the moment about the vertical axis is the only one that can be very different from zero.

Sincerely,

Manoj Srinivasan, Ph.D.

Guest Editor

PLOS Computational Biology

Stefano Allesina

Deputy Editor

PLOS Computational Biology

[LINK]

Both reviewers were mostly satisfied with the revisions, but both reviewers, each, indicate a disagreement or a misunderstanding with respect to one of their comments/suggestions. We look forward to your addressing these issues and clearing up the misunderstandings in the next revision. One of the reviewers also suggests other minor revisions.

I agree with the reviewer that it may make sense to remark on the GRM modifications or comparison, and do not see why the authors say that such comparison "is not meaningful". Also, isn't it is not technically true that feet can only produce moments against the ground in the vertical direction: it's just that the moment about the vertical axis is the only one that can be very different from zero.

Reviewer's Responses to Questions

**Comments to the Authors:**

Reviewer #1: See attachment.

Reviewer #2: Congratualations on a fine piece of work. Please find attached the second review commentary with the final remaining issue.

**Have all data underlying the figures and results presented in the manuscript been provided?**

Reviewer #1: Yes

Reviewer #2: Yes

PLOS authors have the option to publish the peer review history of their article (what does this mean?). If published, this will include your full peer review and any attached files.

Reviewer #1: No

Reviewer #2: **Yes: **Koen K. Lemaire
---

## [Decision Letter · Decision Letter 2]

28 Jan 2021

Dear Dr. Bishop,

Thank you very much for submitting your manuscript "Computational modelling of muscle fibre operating ranges in the hindlimb of a small ground bird (Eudromia elegans), with implications for modelling locomotion in extinct species" for consideration at PLOS Computational Biology. As with all papers reviewed by the journal, your manuscript was reviewed by members of the editorial board and by several independent reviewers. The reviewers appreciated the attention to an important topic. Based on the reviews, we are likely to accept this manuscript for publication, providing that you modify the manuscript according to the review recommendations.

The reviewer has asked for very minor edits/clarifications regarding the GRM calculation and comparisons. Thank you.

Sincerely,

Manoj Srinivasan, Ph.D.

Guest Editor

PLOS Computational Biology

Stefano Allesina

Deputy Editor

PLOS Computational Biology

[LINK]

The reviewer has asked for very minor edits/clarifications regarding the GRM calculation and comparisons. Thank you. As the reviewer asks, it may be worth remarking how GRMs were handled more explicitly. 

Reviewer's Responses to Questions

**Comments to the Authors:**

Reviewer #1: All my comments/questions have been answered. Congratulations on a great paper!

Reviewer #2: Please find the review attached. Best wishes in the new year.

**Have all data underlying the figures and results presented in the manuscript been provided?**

Reviewer #1: Yes

Reviewer #2: Yes

PLOS authors have the option to publish the peer review history of their article (what does this mean?). If published, this will include your full peer review and any attached files.

Reviewer #1: **Yes: **Anne Koelewijn

Reviewer #2: **Yes: **Koen K. Lemaire
---

## [Editor Report · Decision Letter 3]

1 Mar 2021

Dear Mr. Bishop,

We are pleased to inform you that your manuscript 'Computational modelling of muscle fibre operating ranges in the hindlimb of a small ground bird (Eudromia elegans), with implications for modelling locomotion in extinct species' has been provisionally accepted for publication in PLOS Computational Biology.

Best regards,

Manoj Srinivasan, Ph.D.

Guest Editor

PLOS Computational Biology

Stefano Allesina

Deputy Editor

PLOS Computational Biology

Thank you for addressing the reviewer comments.

---

## [Editor Report · Acceptance letter]

18 Mar 2021

PCOMPBIOL-D-20-01173R3 

Computational modelling of muscle fibre operating ranges in the hindlimb of a small ground bird (*Eudromia elegans*), with implications for modelling locomotion in extinct species

Dear Dr Bishop,

I am pleased to inform you that your manuscript has been formally accepted for publication in PLOS Computational Biology. Your manuscript is now with our production department and you will be notified of the publication date in due course.

With kind regards,

Alice Ellingham
